# Dist Loss: Enhancing Regression in Few-Shot Region through Distribution Distance Constraint

**Guangkun Nie**[1,2,*]**, Gongzheng Tang**[1,2,*]**, Shenda Hong**[1,2,3,†]

[1] Institute of Medical Technology, Health Science Center of Peking University, Beijing, China
[2] National Institute of Health Data Science, Peking University, Beijing, China
[3] Institute for Artificial Intelligence, Peking University, Beijing, China
`nieguangkun@stu.pku.edu.cn, gztang@hsc.pku.edu.cn, hongshenda@pku.edu.cn`

## Abstract

Imbalanced data distributions are prevalent in real-world scenarios, presenting significant challenges in both classification and regression tasks. This imbalance often causes deep learning models to overfit in regions with abundant data (many-shot regions) while underperforming in regions with sparse data (few-shot regions). Such characteristics limit the applicability of deep learning models across various domains, notably in healthcare, where rare cases often carry greater clinical significance. While recent studies have highlighted the benefits of incorporating distributional information in imbalanced classification tasks, similar strategies have been largely unexplored in imbalanced regression. To address this gap, we propose Dist Loss, a novel loss function that integrates distributional information into model training by jointly optimizing the distribution distance between model predictions and target labels, alongside sample-wise prediction errors. This dual-objective approach encourages the model to balance its predictions across different label regions, leading to significant improvements in accuracy in few-shot regions. We conduct extensive experiments across three datasets spanning computer vision and healthcare: IMDB-WIKI-DIR, AgeDB-DIR, and ECG-K-DIR. The results demonstrate that Dist Loss effectively mitigates the impact of imbalanced data distributions, achieving state-of-the-art performance in few-shot regions. Furthermore, Dist Loss is easy to integrate and complements existing methods. To facilitate further research, we provide our implementation at `https://github.com/Ngk03/DIR-Dist-Loss`.

## 1 Introduction

Imbalanced data distributions are prevalent in the real world, where certain target values are significantly underrepresented Buda et al. (2018); Liu et al. (2019). In imbalanced regression tasks, deep learning models tend to bias their predictions toward regions with abundant data (many-shot regions) to minimize overall error, resulting in significantly higher errors in sparse-data regions (few-shot regions). This phenomenon severely limits the applicability of deep learning models in certain contexts, such as healthcare scenarios where rare cases often carry significant importance, and significant errors in these samples could lead to potential adverse events.

Taking the prediction of potassium concentration from electrocardiogram (ECG) signals as an example, the model takes ECG signals as input and predicts the corresponding potassium concentration. Figure 1a illustrates the distribution of potassium concentrations in a real-world dataset, where the majority of samples fall within the normal range, while abnormal potassium concentrations ($\leq$ 3.5 mmol/L or $\geq$ 5.5 mmol/L) are rare and concentrated in the few-shot region. Due to this imbalance, deep learning models tend to bias their predictions towards the normal range in order to

---

[*] Authors contributed equally to this research.
[†] Corresponding author.

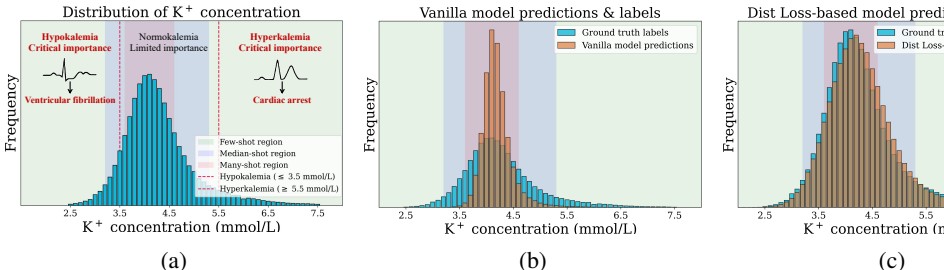

Figure 1: A real-world healthcare task of potassium ($K^+$) concentration regression from ECGs. (a) Both hyperkalemia (high $K^+$) and hypokalemia (low $K^+$) are predominantly found in the few-shot region, with normal $K^+$ are located in the many-shot region. Hyperkalemia and hypokalemia are life-threatening conditions that can lead to cardiac arrest and ventricular fibrillation, necessitating accurate and timely detection. Conversely, normal $K^+$ concentrations (the many-shot region) are of little concern, as inaccurate and untimely detection of these samples has minimal impact. Here, we follow Yang et al. (2021) to define the few-, median-, many-shot regions. (b) illustrates the significant distribution discrepancy between the vanilla model's predictions and the labels, stemming from the imbalanced data distribution. Here, the term "vanilla model" refers to a model that employs no specialized techniques to address imbalanced data. The orange histogram represents the label distribution, while the blue histogram depicts the prediction distribution from the vanilla model. It is evident that the model's predictions are heavily concentrated in the many-shot region and seldom fall into the few-shot region. (c) demonstrates the effectiveness of Dist Loss in reducing the distribution discrepancy. The orange histogram indicates the label distribution, and the blue histogram shows the prediction distribution from the model enhanced with Dist Loss. It is clear that the distribution discrepancy is significantly reduced.

minimize overall error (Figure 1b). However, abnormal potassium concentrations can severely affect metabolism and cardiac function, potentially leading to life-threatening arrhythmias or sudden death Ferreira et al. (2020); Crotti et al. (2020); Kim et al. (2023). Therefore, in clinical settings, accurately predicting abnormal potassium concentrations is far more critical than accurately identifying normal levels Galloway et al. (2019); Harmon et al. (2024), yet this remains a challenge for existing deep learning models. Enhancing model accuracy in the few-shot region under imbalanced data distributions continues to be both a significant challenge and an important goal Branco et al. (2017); Steininger et al. (2021).

In imbalanced regression tasks, a significant challenge lies in the substantial discrepancy between the distribution of the model's predictions and the true label distribution, as illustrated in Figure 1b. The orange histogram represents the ground truth, while the blue histogram depicts the distribution of predictions made by the vanilla model, which refers to a model trained without techniques specifically designed to address data imbalance (using L1 loss here). It is evident that the model predominantly predicts values within the many-shot region, with very few predictions falling within the few-shot region. This discrepancy highlights a critical consequence of imbalanced data distributions: the misalignment between the model's predictions and the target labels. While prior research has focused on addressing the adverse effects of class imbalance in classification tasks by incorporating distributional information into the training process Feng et al. (2018); Zheng et al. (2020); Tian et al. (2020), similar strategies have been largely unexplored in imbalanced regression. Therefore, a key research direction is to investigate whether leveraging distributional information can effectively reduce prediction errors in the few-shot region by aligning the distribution of the model's predictions with the underlying label distribution.

Based on this concept, we propose Dist Loss, a novel loss function comprising two key components: distribution alignment optimization, which minimizes the discrepancy between the model's prediction distribution and the label distribution, and sample-wise prediction optimization, which ensures accurate predictions at the individual sample level. The distribution alignment optimization is achieved through three steps: (1) Generating pseudo-labels: we apply kernel density estimation (KDE) to the label set Parzen (1962) to model the label distribution and sample pseudo-labels that reflect this distribution; (2) Constructing pseudo-predictions: we sort the model's predictions to construct pseudo-predictions that represent the prediction distribution; (3) Measuring distribu-

tion discrepancy: we approximate the discrepancy between the prediction and label distributions by computing the distance between the pseudo-labels and pseudo-predictions. By jointly optimizing distribution alignment and sample-wise prediction accuracy, Dist Loss mitigates errors in individual predictions while ensuring that the model's prediction distribution better conforms to the label distribution. As shown in Figure 1c, this approach effectively addresses the distribution mismatch caused by data imbalance and has been proven to improve prediction accuracy in the few-shot regions.

To validate the effectiveness of Dist Loss, we conduct comprehensive experiments on three datasets spanning computer vision and healthcare: IMDB-WIKI-DIR, AgeDB-DIR, and our meticulously curated ECG-K-DIR dataset. Our results demonstrate a substantial improvement in accuracy for rare samples, leading to state-of-the-art (SOTA) performance. Furthermore, our experiments show that Dist Loss is compatible with existing techniques, yielding further performance gains when integrated.

In summary, the contributions of this paper are:

- We analyze the impact of imbalanced data distributions in regression from a distributional perspective and introduce the concept of aligning the model's prediction distribution with the label distribution by leveraging distributional priors.

- We propose a differentiable approach for measuring distribution distance in regression tasks, extending distribution alignment techniques from classification to regression.

- Extensive experiments on diverse datasets demonstrate the effectiveness of Dist Loss in imbalanced regression, achieving SOTA performance in few-shot regions.

## 2 RELATED WORK

### 2.1 IMBALANCED CLASSIFICATION

Research on the problem of imbalanced classification mainly focuses on improving the loss function to enhance the model's ability to identify the minority class. Weighted cross entropy King & Zeng (2001) gives higher weights to minority class samples, allowing the model to pay more attention to minority class samples when facing class imbalance. Focal loss Lin (2017) reduces the influence of the majority class by dynamically adjusting the weights in the loss function, further improving the performance of the minority class. Combining data augmentation and resampling techniques is also a common strategy. RUSBoost Seiffert et al. (2009) combines random undersampling and boosting to reduce the majority class while maintaining the performance of the model. SMOTE Chawla et al. (2002) further improves the classification results by expanding the minority class data through synthetic samples. The combination of adversarial training and loss functions has also gradually attracted attention, and adversarial reweighting Sagawa et al. (2019) improves the accuracy of minority classes.

### 2.2 IMBALANCED REGRESSION

Unlike classification tasks, where labels are discrete and bounded, regression tasks involve continuous and unbounded labels, and the distances between labels carry meaningful semantic information. These fundamental differences prevent methods designed for imbalanced classification from being directly applied to imbalanced regression. Existing methods for addressing imbalanced regression can be categorized into three levels: input, feature, and model output. At the input level, methods primarily focus on resampling the training dataset. SMOTE Chawla et al. (2002); Torgo et al. (2013) and its variant SMOGN Branco et al. (2017) generate new samples by interpolating between minority samples and their nearest neighbors. Branco et al. (2018) further enhances this approach by integrating a bagging-based ensemble method with SMOTE to mitigate the impact of imbalanced data distributions. At the feature level, Yang et al. (2021) introduces feature distribution smoothing (FDS), which transfers feature statistics between nearby target bins to smooth the feature space. VIR Wang & Wang (2024) borrows data with similar regression labels to compute the variational distribution of the latent representation. Ranksim Gong et al. (2022) employs contrastive learning to bring the feature representations of samples with similar labels closer while pushing apart those with dissimilar labels. Similarly, ConR Keramati et al. (2024) designs positive and negative sample pairs

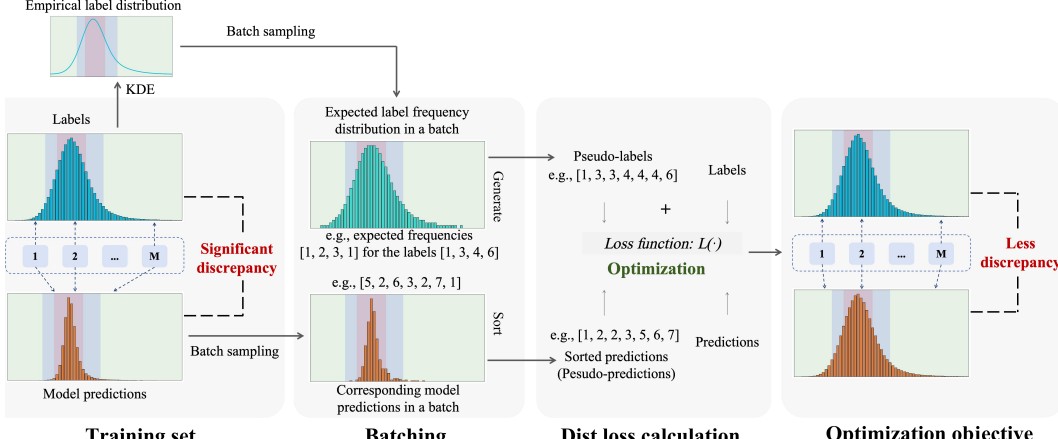

Figure 2: The presence of imbalanced data distributions introduces a noticeable distribution discrepancy between the model's predictions and labels. Dist Loss mitigates this imbalance by simultaneously minimizing this discrepancy and sample-wise prediction errors. Initially, KDE is applied to estimate the label distribution and compute the expected frequency of each label within a batch, thereby generating pseudo-labels that incorporate label distribution information. For example, given the labels [1, 3, 4, 6] and their computed expected frequencies [1, 2, 3, 1], the resulting pseudo-labels would be [1, 3, 3, 4, 4, 4, 6], where each label appears according to its expected frequency. Subsequently, the model's predictions within a batch are sorted to obtain an ordered sequence that captures the prediction distribution. For instance, if the model's initial predictions are [5, 2, 6, 3, 2, 7, 1], sorting them yields [1, 2, 2, 3, 5, 6, 7], preserving the distributional characteristics of the predictions. Measuring the distance between these pseudo-labels and pseudo-predictions, which both encapsulate distribution information, provides an approximation of the distributional discrepancy. By optimizing both the distribution distance and sample-level prediction errors during training, the model effectively alleviates the adverse effects of imbalanced data, significantly enhancing accuracy, particularly in few-shot regions.

based on label similarity, transferring label space relationships to the feature space in a contrastive manner. At the model output and label level, regressor retraining (RRT) Yang et al. (2021) decouples the training of the encoder and regressor, retraining the regressor with inverse reweighting after normal encoder training. DenseLoss Steininger et al. (2021) and label distribution smoothing (LDS) Yang et al. (2021) measure label rarity through KDE, assigning higher weights to rare samples to enhance the model's focus on minority samples. Balanced MSE Ren et al. (2022) leverages the training label distribution prior to restore balanced predictions.

However, existing research on imbalanced regression often overlooks the significant distribution discrepancy between model predictions and labels, and distribution information, which has been proven effective in imbalanced classification, is rarely utilized. In contrast, our approach introduces distribution distance optimization on top of the traditional focus on sample-wise prediction error. By concurrently aligning the prediction distribution with the label distribution and minimizing sample-level prediction errors, our method significantly improves accuracy in few-shot regions. This enhancement is achieved without incurring additional computational costs or requiring meticulous hyperparameter tuning. Extensive experiments demonstrate the superiority of our approach in handling critical and informative rare samples in few-shot regions, achieving SOTA results.

## 3 METHOD

### 3.1 PROBLEM SETTING

Let $\mathcal{D}$ be a training dataset consisting of $N$ samples, denoted as $\mathcal{D} = \{(\mathbf{x}_{(i)}, y_{(i)})\}_{i=1}^{N}$, where $\mathbf{x}_{(i)} \in \mathbb{R}^d$ represents the input and $y_{(i)} \in \mathbb{R}$ denotes the corresponding label. To facilitate processing, the continuous label space $\mathcal{Y}$ is discretized into $B$ bins of equal width, such that $\mathcal{Y} = \bigcup_{b=1}^{B}[y_b, y_{b+1})$,

where $y_b$ is the lower bound of bin $b$, and $y_1 < y_2 < \cdots < y_B$. In subsequent discussions, for convenience, the lower bound $y_b$ of the bin $[y_b, y_{b+1})$ will represent any label value $y_{(i)}$ that falls within that bin. The set of bins is denoted as $\mathcal{B} = \{1, 2, \ldots, B\}$. In practical scenarios, the width of each bin, denoted $\Delta_y$, indicates the minimum resolution of interest when processing the label space. For example, in age estimation, one might set the bin width to 1, resulting in $\Delta_y = y_{b+1} - y_b = 1, \quad \forall b \in \mathcal{B}$. Additionally, we define the probability of observing a label $y_i$ as $p_i$, and the probability distribution of labels can be estimated using KDE, which is a non-parametric statistical method used to estimate the probability density function of a random variable without assuming a specific distribution form. Notably, the discretization of the label space serves to facilitate the use of label distribution information while preserving the regression nature of the problem, rather than transforming it into a classification task. Since continuous probability density functions are difficult to process directly, we discretize them into bins for practical computation and estimation. This discretization applies to the entire label space, which includes all possible values, not just those occurring in the training dataset.

## 3.2 DIST LOSS

One of the optimization objectives of Dist Loss is to minimize the distance between the prediction and label distributions in regression tasks. The core challenge lies in measuring the distance between these two distributions in a differentiable manner. Traditional metrics for measuring distribution distance, such as Kullback-Leibler divergence and Jensen-Shannon divergence, cannot be directly implemented in a differentiable form for regression tasks. Therefore, we have devised an alternative approach in the implementation of Dist Loss to realize a differentiable distribution distance measurement in regression scenarios. Specifically, we approximate the distance between the label and prediction distributions by sampling from these distributions and quantifying the differences between the sampled values to estimate the distance.

### 3.2.1 CALCULATION OF DIST LOSS

As illustrated in Figure 2, we sample from the label and prediction distributions to generate pseudo-labels and pseudo-predictions, which encapsulate the distribution information of the labels and predictions. Taking the generation of pseudo-labels as an example, we will now detail the process.

To generate pseudo-labels that contain label distribution information, we first randomly sample $M$ points from the label distribution, where M corresponds to the batch size during model training. The expected frequencies of the label $y_i$ can be estimated by multiplying the number of sampling points $M$ by the probability of that label $p_i$. Based on this, we construct a sequence $\mathcal{N}_L = (n_1, n_2, \cdots, n_B)$ to represent these expected frequencies, where $n_i = M \cdot p_i$. Each element in the obtained $\mathcal{N}_L$ represents the expected frequencies of the corresponding label. Since these frequencies may be fractional, we need to convert them to integers while ensuring that the sum after conversion still equals $M$. Here, we denote the converted integer sequence by $\mathcal{N}_{L'} = (n_1', n_2', \cdots, n_B')$. To acquire $\mathcal{N}_{L'}$, we first take the floor of each element in $\mathcal{N}_L$ to obtain the sequence $\mathcal{N}_{L_f} = (\lfloor n_1 \rfloor, \lfloor n_2 \rfloor, \cdots, \lfloor n_B \rfloor)$. Then we calculate the difference $a$, which represents the difference between the sum of the original expected frequencies ($M$) and the sum after applying the floor function, following $a = M - \sum_{i=1}^{B} \lfloor n_i \rfloor$. Using the difference $a$, we construct an auxiliary sequence $\mathcal{A}$, which determines how to distribute the difference to the elements of $\mathcal{N}_{L_f}$ to ensure the sum is $M$:

$$a_i = \begin{cases} 1, & \text{if } i \le \lfloor \frac{a+1}{2} \rfloor \text{ or } i > B - \lfloor \frac{a}{2} \rfloor \\ 0, & \text{otherwise} \end{cases}, \tag{1}$$

Each $n_i'$ is determined by adding $a_i$ to the corresponding element in $\mathcal{N}_{L_f}$, where $n_i' = \lfloor n_i \rfloor + a_i$, and $i \in \mathcal{B}$. Finally, we generate the corresponding pseudo-labels $\mathcal{S}_L$ based on the expected frequencies, here each element $S_{L_j}$ is represented as:

$$S_{L_j} = y_{\arg\min_{i \in \mathcal{B}}\left(\sum_{k=1}^{i} n_k \ge j\right)}, \quad \text{for } j = 1, \ldots, M, \tag{2}$$

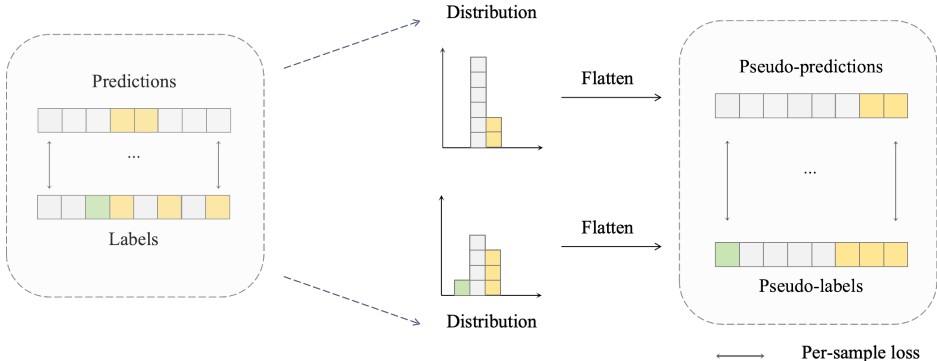

Figure 3: To illustrate the core concept behind Dist Loss, the figure simplifies its computation by assuming that the batch size equals the total number of training samples.

where $j = 1, \ldots, M$ indexes the position in the sequence. To illustrate with a concrete example, consider a label sequence $(y_1, y_2, y_3) = (4, 5, 6)$ and an obtained frequency sequence $\mathcal{N}_{L'} = (1, 2, 3)$. In this case, the generated pseudo-labels $S_L$ would be $(y_1, y_2, y_2, y_3, y_3, y_3)$, corresponding to the sequence $(4, 5, 5, 6, 6, 6)$.

Similarly, we can perform $M$-point sampling on the prediction distribution and apply the same procedure to obtain the pseudo-predictions $\mathcal{S}_P$, which capture the characteristics of the prediction distribution. However, in practice, this process can be simplified by sorting the model predictions within a batch. It is worth noting that although pseudo-labels can also be generated by sorting the labels within a batch, we choose not to adopt this approach here. Instead, we aim to better leverage the full label distribution, as the batch serves merely as a sample. To explain why sorted sequences capture distribution information, the pseudo-labels $\mathcal{S}_L$ and pseudo-predictions $\mathcal{S}_P$ can be viewed as one-dimensional representations of the label and prediction distributions, respectively (Figure 3).

By measuring the distance between the pseudo-predictions and pseudo-labels, we can approximate the distance between their respective distributions. Let $L(\cdot)$ be a function that measures the distance between two sequences; then, the distribution distance can be expressed as $L(\mathcal{S}_P, \mathcal{S}_L)$. Furthermore, using the function $L(\cdot)$, we can simultaneously evaluate the sample-wise prediction errors, which measure the discrepancy between individual predicted values and their corresponding ground truth labels. Specifically, let $Y_{\text{batch}}$ and $\hat{Y}_{\text{batch}}$ denote the sets of ground truth labels and model predictions in a batch, respectively. The sample-wise prediction error can then be formulated as $L(Y_{\text{batch}}, \hat{Y}_{\text{batch}})$. Ultimately, by jointly optimizing both the distributional distance and the sample-level prediction errors during training, we can mitigate the issue of distribution mismatch without compromising overall accuracy, thus addressing the challenges posed by imbalanced data distributions.

### 3.2.2 FAST DIFFERENTIABLE SORTING

As previously mentioned, the obtained pseudo-predictions are in ascending order, whereas the order of the model's actual predictions is random in practical scenarios. Therefore, it is necessary to sort the model's predictions to obtain the pseudo-predictions. Since the sorting operation is non-differentiable, we employ a fast differentiable sorting algorithm Blondel et al. (2020) to ensure the differentiability of the entire computation process.

This method achieves the sorting operation by defining it as projections on permutation polytopes. Specifically, for any given vector $w \in \mathbb{R}^n$, we construct the permutation polytope $P(w)$, which represents the convex hull of all possible permutations of $w$, i.e.,

$$P(w) := \text{conv}(\{w_\sigma : \sigma \in \Sigma\}), \tag{3}$$

where $\Sigma$ denotes all permutations of $[n]$. The sorting operation $s(\theta)$ is defined as the solution to the linear programming problem that maximizes the dot product with $\rho$ (a strictly decreasing vector) on $P(\theta)$, i.e.,

$$s(\theta) = \arg \max_{y \in P(\theta)} \langle y, \rho \rangle. \tag{4}$$

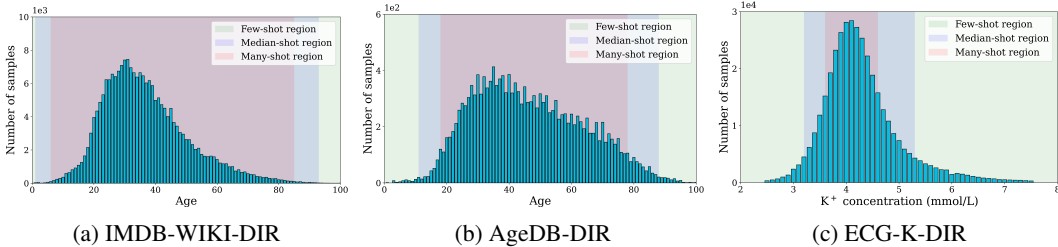

(a) IMDB-WIKI-DIR           (b) AgeDB-DIR           (c) ECG-K-DIR

Figure 4: Overview of label distributions in the training sets for the IMDB-WIKI-DIR, AgeDB-DIR, and ECG-K-DIR datasets. The classification of shot types for IMDB-WIKI-DIR and AgeDB-DIR follows the definitions provided in Yang et al. (2021).

To ensure the differentiability of the sorting operation, a regularization term $\Psi$ is introduced, transforming the sorting operation into tractable projection problems:

$$P_\Psi(z, w) = \arg \min_{\mu \in P(w)} \left\{ \frac{1}{2} \|\mu - z\|^2 + \Psi(\mu) \right\}, \tag{5}$$

where $\Psi$ is a strongly convex function, ensuring the differentiability of the problem. This approach enables forward propagation with $O(n \log n)$ time complexity and backward propagation with $O(n)$ time complexity.

## 4 EXPERIMENTS

### 4.1 BENCHMARKS AND BASELINES

We evaluated our method on three datasets, focusing on tasks of age estimation and potassium concentration prediction. The IMDI-WIKI-DIR dataset Yang et al. (2021), derived from the IMDB-WIKI dataset Rothe et al. (2018), consists of 213,553 facial image pairs annotated with age information. This dataset is partitioned into 191,509 samples for training, 11,022 for validation, and 11,022 for testing. The AgeDB-DIR dataset Yang et al. (2021), derived from the AgeDB dataset Moschoglou et al. (2017), comprises 16,488 facial image pairs with age annotations. It is divided into 12,208 samples for training, 2,140 for validation, and 2,140 for testing. The ECG-K-DIR dataset, sourced from the MIMIC-IV dataset Johnson et al. (2020), includes 375,745 pairs of single-lead ECG signals paired with potassium concentration values. This dataset is divided into 365,549 samples for training, 5,098 for validation, and 5,098 for testing. All these datasets are characterized by imbalanced training sets and balanced validation and test sets. The label distributions of these three datasets are shown in Figure 4. Please refer to Appendix A.1 and A.2 for baseline and implementation details.

### 4.2 EVALUATION METRICS

Following the evaluation metrics of Yang et al. (2021), we report the results for four shots: all, many, median, and few, where all represents the entire dataset, and many/median/few correspond to areas of high/medium/low sample density within the dataset. For the IMDB-WIKI-IR and AgeDB-DIR datasets, we maintain consistency with previous studies, where few/median/many correspond to areas with fewer than 20, between 20-100, and more than 100 samples, respectively. For the ECG-K-DIR dataset, assuming that the maximum number of samples for a single label is $n_{max}$, we define areas with more than 0.5 $n_{max}$, between 0.15-0.5 $n_{max}$, and fewer than 0.15 $n_{max}$ samples as many/median/few shots areas, respectively. For each dataset, we report the mean absolute error (MAE) and the geometric mean (GM).

### 4.3 MAIN RESULTS

Table 1 presents the results of baselines and our method in the few-shot region across three datasets, along with a comparison of these results. For detailed results on each dataset, please refer to Appendix A.3. This table is divided into two sections. The first section displays the results of the

Table 1: Results are presented for the few-shot region on the IMDB-WIKI-DIR, AgeDB-DIR, and ECG-K-DIR datasets. The first section of the table reports the results of baselines and our method, with the best results highlighted in bold and red. In the second section, improvements over corresponding baselines are reported in bold and green, while decreases in performance are reported in bold and blue.

| | MAE | | | GM | | |
|---|---|---|---|---|---|---|
| | IMDB-WIKI-DIR | AgeDB-DIR | ECG-K-DIR | IMDB-WIKI-DIR | AgeDB-DIR | ECG-K-DIR |
| Vanilla | 26.930 | 12.894 | 1.771 | 21.254 | 9.789 | 1.578 |
| + LDS | 22.753 | 11.279 | 1.510 | 12.803 | 7.846 | 1.190 |
| + FDS | 24.908 | 11.161 | 1.737 | 14.361 | 7.361 | 1.529 |
| + Ranksim | 25.999 | 12.569 | 1.791 | 19.690 | 9.495 | 1.600 |
| + ConR | 25.408 | 12.623 | 1.756 | 17.022 | 8.787 | 1.556 |
| + Balanced MSE | 23.542 | 9.613 | 1.417 | **12.603** | 6.248 | 1.046 |
| **+ Dist Loss (Ours)** | **22.550** | **9.122** | **1.329** | 14.288 | **5.453** | **0.978** |
| Ours vs. Vanilla | **+ 4.380** | **+ 3.772** | **+ 0.442** | **+ 6.966** | **+ 4.336** | **+ 0.600** |
| Ours vs. LDS | **+ 0.203** | **+ 2.157** | **+ 0.181** | **- 1.485** | **+ 2.393** | **+ 0.212** |
| Ours vs. FDS | **+ 2.358** | **+ 2.039** | **+ 0.408** | **+ 0.073** | **+ 1.908** | **+ 0.551** |
| Ours vs. Ranksim | **+ 3.449** | **+ 3.447** | **+ 0.462** | **+ 5.402** | **+ 4.042** | **+ 0.622** |
| Ours vs. ConR | **+ 2.858** | **+ 3.501** | **+ 0.427** | **+ 2.734** | **+ 3.334** | **+ 0.578** |
| Ours vs. Balanced MSE | **+ 0.992** | **+ 0.491** | **+ 0.088** | **- 1.685** | **+ 0.795** | **+ 0.068** |

Table 2: Results are presented for the few-shot region on the IMDB-WIKI-DIR, AgeDB-DIR, and ECG-K-DIR datasets. Each section of the table reports the results of a baseline and the baseline incorporating our method, with the better results highlighted in bold.

| | MAE | | | GM | | |
|---|---|---|---|---|---|---|
| | IMDB-WIKI-DIR | AgeDB-DIR | ECG-K-DIR | IMDB-WIKI-DIR | AgeDB-DIR | ECG-K-DIR |
| + LDS | 22.753 | 11.279 | 1.510 | **12.803** | 7.846 | 1.190 |
| + LDS + **Dist Loss** | **22.331** | **10.437** | **1.325** | 13.021 | **7.051** | **0.957** |
| + FDS | 24.908 | 11.161 | 1.737 | **14.361** | 7.361 | 1.529 |
| + FDS + **Dist Loss** | **24.112** | **10.444** | **1.428** | 14.929 | **6.696** | **1.099** |
| + Ranksim | 25.999 | 12.569 | 1.791 | 19.690 | 9.495 | 1.600 |
| + Ranksim + **Dist Loss** | **23.772** | **12.102** | **1.325** | **15.422** | **8.515** | **0.970** |
| + ConR | 25.408 | 12.623 | 1.756 | 17.022 | **8.787** | 1.556 |
| + ConR + **Dist Loss** | **22.700** | **12.303** | **1.336** | **14.713** | 9.123 | **0.987** |
| + Balanced MSE | 23.542 | 9.613 | 1.417 | **12.603** | 6.248 | 1.046 |
| + Balanced MSE + **Dist Loss** | **22.597** | **9.110** | **1.357** | 14.238 | **5.585** | **0.996** |

baselines and our method, with the best results highlighted in bold and red. The second section shows the improvement of our method over each baseline, with green bold indicating superior performance of our method and blue bold indicating otherwise. From the first section of the table, it is evident that our method achieves the best results in five out of six metrics across the three datasets, with SOTA performances of 22.550, 9.122, and 1.329 on the IMDB-WIKI-DIR, AgeDB-DIR, and ECG-K-DIR datasets, respectively. The second section reveals that our method outperforms in 28 out of 30 metrics. Notably, compared to Balanced MSE, which also involves fine-tuning the linear layers of a pre-trained model and employs data distribution priors, our method demonstrates superior performance in the few-shot region, highlighting the effectiveness of our approach.

Table 2 further illustrates the complementary nature of our method with existing approaches. This table is divided into five sections, each showcasing the results of one baseline and the combined results with our method, with the best results within each section highlighted in bold and black. From this table, it is shown that our method achieves better results in 26 out of 30 metrics. Taking the MAE metric as an example, incorporating our method leads to improved performance in the few-shot region across all three datasets, achieving the best results of 22.331, 9.110, and 1.325 on the IMDB-WIKI-DIR, AgeDB-DIR, and ECG-K-DIR datasets, respectively. These experimental results demonstrate a key advantage of our method, namely its ability to effectively complement existing methods, thereby enhancing model performance in the few-shot region.

Table 3: Time consumption (in seconds) of one training epoch for the IMDB-WIKI-DIR, AgeDB-DIR, and ECG-K-DIR datasets, with batch sizes of 64, 64, and 256, respectively.

| | IMDB-WIKI-DIR | AgeDB-DIR | ECG-K-DIR |
|---|---|---|---|
| Vanilla | 399.8 | 31.8 | 94.6 |
| + LDS | 401.2 | 31.4 | 104.0 |
| + FDS | 567.5 | 43.6 | 155.1 |
| + Ranksim | 512.6 | 40.2 | 135.1 |
| + ConR | 1168.7 | 91.6 | 192.1 |
| + Balanced MSE | 152.6 | 14.2 | 51.8 |
| **+ Dist Loss (Ours)** | 154.0 | 15.1 | 58.7 |

Table 4: Ablation study on loss functions measuring sequence difference. $L_1$ represents MAE Loss, $L_2$ represents MSE Loss, $INV-$ denotes the the version of these loss functions that are inverse probability weighted. Results on the few-shot region are reported, with the best results in each section are in bold.

| | MAE | | | GM | | |
|---|---|---|---|---|---|---|
| | IMDB-WIKI-DIR | AgeDB-DIR | ECG-K-DIR | IMDB-WIKI-DIR | AgeDB-DIR | ECG-K-DIR |
| Vanilla | 26.930 | 12.894 | 1.771 | 21.254 | 9.789 | 1.578 |
| + Dist Loss ($INV-L_1$) | 23.334 | 9.802 | 1.467 | 15.437 | 6.298 | 1.044 |
| + Dist Loss ($INV-L_2$) | **22.516** | **9.122** | **1.329** | **13.752** | **5.453** | **0.978** |
| + LDS | 22.753 | 11.279 | 1.510 | 12.803 | 7.846 | 1.190 |
| + Dist Loss ($INV-L_1$) | **22.178** | **9.872** | 1.413 | **11.334** | **6.109** | 0.984 |
| + Dist Loss ($INV-L_2$) | 22.331 | 10.437 | **1.325** | 13.021 | 7.051 | **0.957** |
| + FDS | 24.908 | 11.161 | 1.737 | **14.361** | 7.361 | 1.529 |
| + Dist Loss ($INV-L_1$) | **23.692** | **9.969** | 1.515 | 14.399 | **6.026** | 1.122 |
| + Dist Loss ($INV-L_2$) | 24.112 | 10.444 | **1.428** | 14.929 | 6.696 | **1.099** |
| + Ranksim | 25.999 | 12.569 | 1.791 | 19.690 | 9.495 | 1.600 |
| + Dist Loss ($INV-L_1$) | 23.894 | **11.877** | 1.577 | 16.036 | **8.164** | 1.330 |
| + Dist Loss ($INV-L_2$) | **23.772** | 12.102 | **1.325** | **15.422** | 8.515 | **0.970** |
| + ConR | 25.408 | 12.623 | 1.756 | 17.022 | 8.787 | 1.556 |
| + Dist Loss ($INV-L_1$) | 23.281 | **11.948** | 1.452 | 15.586 | **8.605** | 1.044 |
| + Dist Loss ($INV-L_2$) | **22.700** | 12.303 | **1.336** | **14.713** | 9.123 | **0.987** |
| + Balanced MSE | 23.542 | 9.613 | 1.417 | **12.603** | 6.248 | 1.046 |
| + Dist Loss ($INV-L_1$) | 23.539 | 9.762 | 1.474 | 15.000 | 6.198 | 1.051 |
| + Dist Loss ($INV-L_2$) | **22.597** | **9.110** | **1.357** | 14.238 | **5.585** | **0.996** |

## 4.4 TIME CONSUMPTION ANALYSIS

Table 3 presents the time required to train each method for one epoch on the IMDB-WIKI-DIR, AgeDB-DIR, and ECG-K-DIR datasets, with all times reported in seconds. It can be observed that Balanced MSE and Dist Loss have the shortest training times, attributed to their approach of fine-tuning the model's linear layers. The time consumption of LDS and the vanilla model are largely consistent, as these methods only weight the loss function without significantly increasing computational load. For methods operating at the feature level, including FDS, Ranksim, and ConR, a notable increase in model training time is evident, due to the computational intensity associated with feature-level operations.

## 4.5 ABLATIONS AND ANALYSIS

### 4.5.1 DIFFERENT LOSS FUNCTIONS FOR SEQUENCE DIFFERENCE MEASUREMENT

Dist Loss employs the loss function $L(\cdot)$ to measure the difference between two sequences. In this ablation study, we demonstrate the effects of using different functions, including the inverse probability weighted MAE and MSE losses. The experimental results are shown in Table 4, with detailed results on each dataset provided in Appendix A.4.2. The table illustrates that Dist Loss reliably improves model accuracy in the few-shot region.

Table 5: Ablation study on batch sizes for Dist Loss. Results on the few-shot region are reported.

|  | MAE | | GM | |
| --- | --- | --- | --- | --- |
|  | IMDB-WIKI-DIR | | AgeDB-DIR | |
| 256 | 22.323 | 9.013 | 13.787 | 5.632 |
| 512 | 22.516 | 9.122 | 13.752 | 5.453 |
| 768 | 22.550 | 9.148 | 14.288 | 5.223 |

### 4.5.2 DIFFERENT BATCH SIZES FOR DISTRIBUTION DISTANCE APPROXIMATION

Dist Loss estimates the overall distribution distance between predictions and labels by measuring batch-wise distances during training. This ablation study evaluates the sensitivity of model accuracy to batch size, as detailed in Table 5. We examined batch sizes of 256, 512, and 768, adopting 256 as a standard based on prior research Yang et al. (2021); Gong et al. (2022). The findings show negligible variations in performance with different batch sizes. This could be attributed to the fact that accurately reflecting the distribution information during sampling is more important than requiring each value in the generated pseudo-sequence to be perfectly accurate, which in turn allows for a smaller batch size.

### 4.5.3 DIST LOSS SURPASSES EXISTING METHODS IN THE MEDIAN-SHOT REGION

As depicted in the supplementary Tables 6, 7, and 8 within Appendix A.3, Dist Loss delivers SOTA results, excelling not only in few-shot regions but also in median-shot regions. In our comparison with current methods, Dist Loss achieved the lowest MAE and the second-lowest GM on the IMDB-WIKI-DIR and AgeDB-DIR datasets, with scores of 12.614/7.686 and 7.315/4.563, respectively. Similarly, on the ECG-K-DIR dataset, it secured the highest GM and the second-lowest MAE, recording 0.445 and 0.674, respectively. Moreover, our experiments show that integrating Dist Loss with existing methods consistently improved performance in median-shot regions when measured by both MAE and GM, surpassing the results of using those methods alone on IMDB-WIKI-DIR and AgeDB-DIR datasets. On the ECG-K-DIR dataset, this integration notably increased the GM. In conclusion, these findings validate Dist Loss's efficacy in enhancing model accuracy in both few-shot and median-shot regions.

## 5 CONCLUSION

In this study, we address the significant escalation of prediction errors in few-shot regions, a prevalent challenge in imbalanced regression. By leveraging distribution priors, we introduce a novel loss function, Dist Loss, designed to align the model's prediction distribution with the label distribution throughout the training process. Our extensive experimental evaluation demonstrates that Dist Loss effectively enhances prediction accuracy in few-shot regions, achieving state-of-the-art performance. Furthermore, our results indicate that Dist Loss can be seamlessly integrated with existing methods to further augment their efficacy. We hope our work underscores the critical role of integrating distribution information in tackling deep imbalanced regression tasks.

## ACKNOWLEDGEMENT

This work was supported by the Beijing Natural Science Foundations (QY23040); National Natural Science Foundation of China (62102008); Clinical Medicine Plus X - Young Scholars Project of Peking University, the Fundamental Research Funds for the Central Universities (PKU2024LCXQ030); PKU-OPPO Fund (B0202301);

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

# A APPENDIX

## A.1 BASELINES

To ensure a fair comparison, we followed the experimental setup of Yang et al. (2021) on the IMDB-WIKI-DIR and AgeDB-DIR datasets, i.e., using ResNet-50 as the network architecture and training for 90 epochs. For the ECG-K-DIR dataset, we employed the ResNet variant Net1D Hong et al. (2020) as the network architecture. Given that previous work Yang et al. (2021); Ren et al. (2022); Gong et al. (2022); Keramati et al. (2024) has demonstrated superior performance over loss reweighting and RRT in deep imbalanced regression tasks, we do not include these methods as baselines in this paper. Instead, we focused on widely recognized approaches in the field: LDS, FDS Yang et al. (2021), Ranksim Gong et al. (2022), ConR Keramati et al. (2024), and Balanced MSE Ren et al. (2022). LDS and FDS encourage local similarities in label and feature space, while Ranksim and ConR leverage contrastive learning to translate label similarities into the feature space. Balanced MSE, based on label distribution priors, restores a balanced distribution from an imbalanced dataset. Our experimental findings indicate that not only does our method achieve SOTA performance in few-shot regions, but it also enhances existing methods, offering a complementary strategy to boost their efficacy.

## A.2 IMPLEMENTATION DETAILS

We trained all models on the IMDB-WIKI-DIR and AgeDB-DIR datasets using a single NVIDIA GeForce RTX 3090 GPU and on the ECG-K-DIR dataset using a single NVIDIA GeForce RTX 4090 GPU. To ensure a fair comparison, we followed the training, validation, and test set divisions from Yang et al. (2021) for the IMDB-WIKI-DIR and AgeDB datasets. During training with Dist Loss, we used the same strategy as Balanced MSE, fine-tuning the linear layer based on pre-trained model (vanilla model) parameters. This approach integrates our method with existing methods, using their model parameters as the starting point for fine-tuning. Additionally, we used inverse probability weighted MSE to measure sequence difference in Dist Loss for all datasets, setting the distribution loss component weight to 1.

### A.2.1 IMDB-WIKI-DIR

On the IMDB-WIKI-DIR dataset, we selected ResNet-50 as the network architecture. During training, the training epochs were set to 90, with an initial learning rate of 0.001, which was reduced to 1/10 of its value at the 60th and 80th epochs. We employed the Adam optimizer with a momentum of 0.9 and a weight decay of 0.0001. For our method and Balanced MSE, we used a batch size of 512. For the other baselines, we followed the experimental setups from their original papers. It should be noted that the original training epochs for ConR was 120, which we adjusted to 90 in our experiments to ensure a fair comparison.

### A.2.2 AGEDB-DIR

On the AgeDB dataset, we employed ResNet-50 architecture for our model. The training consisted of 90 epochs with an initial learning rate of 0.001, which was reduced to 1/10 of its original value at the 60th and 80th epochs. We utilized the Adam optimizer with a momentum of 0.9 and a weight decay of 0.0001. For our method and Balanced MSE, we used a batch size of 512. For the other baselines, we followed the experimental configurations outlined in their respective original papers. To ensure a fair comparison, we also set the training epochs for Ranksim and ConR to 90.

### A.2.3 ECG-K-DIR

On the ECG-K-DIR dataset, we utilized the ResNet variant, Net1D Hong et al. (2020), as our network architecture. The training was set for 10 epochs with an initial learning rate of 0.001, which was reduced to 1/10 of its initial value at the 5th and 8th epochs. We employed the Adam optimizer with a momentum of 0.9 and a weight decay of 0.00001. A batch size of 512 was used for all methods. Additionally, for ConR, we constructed positive and negative sample pairs by adding Gaussian noise.

Table 6: Comprehensive results on the IMDB-WIKI-DIR dataset are presented. The table highlights the best results in each section using bold font. Additionally, the best result in each column is indicated in bold and red.

| | MAE | | | | GM | | | |
|---|---|---|---|---|---|---|---|---|
| | All | Many | Median | Few | All | Many | Median | Few |
| Vanilla | 8.143 | **7.260** | 15.758 | 26.930 | 4.642 | **4.211** | 11.522 | 21.254 |
| + **Dist Loss** | **8.028** | 7.461 | **12.614** | **22.516** | **4.593** | 4.335 | **7.686** | **13.752** |
| + LDS | 8.036 | **7.445** | 12.869 | 22.753 | **4.570** | **4.322** | 7.528 | **12.803** |
| + **Dist Loss** | **8.017** | 7.479 | **12.304** | **22.331** | 4.593 | 4.369 | **7.078** | 13.021 |
| + FDS | **7.954** | **7.272** | 13.523 | 24.908 | **4.499** | **4.192** | 8.633 | **14.361** |
| + **Dist Loss** | 8.712 | 8.163 | **12.979** | **24.112** | 5.222 | 4.995 | **7.575** | 14.929 |
| + Ranksim | 7.764 | **6.956** | 14.606 | 25.999 | **4.371** | **3.996** | 9.964 | 19.690 |
| + Dist Loss | **7.721** | 7.129 | **12.401** | **23.772** | 4.422 | 4.183 | **7.091** | **15.422** |
| + ConR | **7.842** | **7.033** | 14.772 | 25.408 | **4.329** | **3.951** | 10.250 | 17.022 |
| + Dist Loss | 7.957 | 7.355 | **12.906** | **22.700** | 4.529 | 4.244 | **8.131** | **14.713** |
| Balanced MSE | **8.033** | **7.441** | 12.768 | 23.542 | 4.716 | 4.450 | 8.035 | **12.603** |
| + Dist Loss | 8.075 | 7.511 | **12.625** | **22.597** | **4.616** | **4.354** | **7.754** | 14.238 |

Table 7: Comprehensive results on the AgeDB-DIR dataset are presented. The table highlights the best results in each section using bold font. Additionally, the best result in each column is indicated in bold and red.

| | MAE | | | | GM | | | |
|---|---|---|---|---|---|---|---|---|
| | All | Many | Median | Few | All | Many | Median | Few |
| Vanilla | **7.506** | **6.558** | 8.794 | 12.894 | 4.798 | **4.176** | 5.957 | 9.789 |
| + Dist Loss | 7.637 | 7.574 | **7.315** | **9.122** | **4.756** | 4.745 | **4.563** | **5.453** |
| + LDS | **7.783** | **7.070** | 8.957 | 11.279 | 5.088 | **4.599** | 6.142 | 7.846 |
| + Dist Loss | 7.810 | 7.341 | **8.464** | **10.437** | **5.043** | 4.752 | **5.474** | **7.051** |
| + FDS | 7.818 | **7.103** | 9.051 | 11.161 | 4.961 | **4.487** | 6.064 | 7.361 |
| + Dist Loss | **7.799** | 7.351 | **8.374** | **10.444** | **4.863** | 4.615 | **5.181** | **6.696** |
| + Ranksim | 7.272 | **6.363** | 8.458 | 12.569 | **4.617** | **3.939** | 6.120 | 9.495 |
| + Dist Loss | **7.234** | 6.506 | **7.960** | **12.102** | 4.629 | 4.097 | **5.637** | **8.515** |
| + ConR | **7.322** | **6.429** | 8.456 | 12.623 | **4.646** | **4.052** | 5.890 | **8.787** |
| + Dist Loss | 7.383 | 6.572 | **8.373** | **12.303** | 4.657 | 4.112 | **5.591** | 9.123 |
| Balanced MSE | 7.663 | **7.540** | 7.353 | 9.613 | **4.658** | **4.558** | 4.511 | 6.248 |
| + Dist Loss | **7.633** | 7.578 | **7.288** | **9.110** | 4.718 | 4.698 | **4.505** | **5.585** |

Table 8: Comprehensive results on the ECG-K-DIR dataset are presented. The table highlights the best results in each section using bold font. Additionally, the best result in each column is indicated in bold and red.

| | MAE | | | | GM | | | |
|---|---|---|---|---|---|---|---|---|
| | All | Many | Median | Few | All | Many | Median | Few |
| Vanilla | 1.235 | **0.274** | 0.685 | 1.771 | 0.835 | **0.193** | 0.622 | 1.578 |
| + Dist Loss | **1.044** | 0.606 | **0.674** | **1.329** | **0.692** | 0.403 | **0.445** | **0.978** |
| + LDS | 1.092 | **0.368** | **0.638** | 1.510 | 0.708 | **0.236** | 0.500 | 1.190 |
| + Dist Loss | **1.031** | 0.557 | 0.671 | **1.325** | **0.671** | 0.363 | **0.455** | **0.957** |
| + FDS | 1.223 | **0.317** | **0.681** | 1.737 | 0.828 | **0.201** | 0.588 | 1.529 |
| + Dist Loss | **1.095** | 0.557 | 0.688 | **1.428** | **0.744** | 0.375 | **0.490** | **1.099** |
| + Ranksim | 1.249 | **0.275** | 0.696 | 1.791 | 0.841 | **0.190** | 0.629 | 1.600 |
| + Dist Loss | **1.040** | 0.587 | **0.683** | **1.325** | **0.692** | 0.381 | **0.487** | **0.970** |
| + ConR | 1.227 | **0.277** | 0.690 | 1.756 | 0.824 | **0.189** | 0.620 | 1.556 |
| + Dist Loss | **1.045** | 0.581 | **0.684** | 11.336 | **0.696** | 0.376 | **0.480** | **0.987** |
| Balanced MSE | 1.106 | 0.606 | 0.727 | 1.417 | 0.722 | 0.383 | 0.475 | 1.046 |
| + Dist Loss | **1.046** | **0.553** | **0.658** | **1.357** | **0.685** | **0.358** | **0.454** | **0.996** |

Table 9: Ablation study examining the impact of batch size on model performance across the IMDB-WIKI-DIR and AgeDB-DIR datasets.

| Dataset | Batch size | MAE | | | | GM | | | |
|---|---|---|---|---|---|---|---|---|---|
| | | All | Many | Median | Few | All | Many | Median | Few |
| IMDB-WIKI-DIR | 256 | 8.072 | 7.514 | 12.591 | 22.323 | 4.603 | 4.340 | 7.808 | 13.787 |
| | 512 | 8.028 | 7.461 | 12.614 | 22.516 | 4.593 | 4.335 | 7.686 | 13.752 |
| | 768 | 7.989 | 7.413 | 12.663 | 22.550 | 4.572 | 4.308 | 7.763 | 14.288 |
| AgeDB-DIR | 256 | 7.668 | 7.638 | 7.281 | 9.013 | 4.741 | 4.768 | 4.370 | 5.632 |
| | 512 | 7.637 | 7.574 | 7.315 | 9.122 | 4.756 | 4.745 | 4.563 | 5.453 |
| | 768 | 7.545 | 7.607 | 7.260 | 9.148 | 4.723 | 4.746 | 4.483 | 5.223 |

### A.3 Comprehensive experimental results

Tables 6, 7, and 8 present a comprehensive overview of our experimental results on the IMDB-WIKI-DIR, AgeDB-DIR, and ECG-K-DIR datasets. The results indicate that our method achieves improvements in model performance on median-shot and few-shot regions without compromising overall error rates. This further demonstrates the effectiveness of our method in sparse data regions.

### A.4 Ablations and analysis

#### A.4.1 Different batch sizes for distribution distance approximation

Table 9 illustrates the impact of varying batch sizes on the final performance across IMDB-WIKI-DIR and AgeDB-DIR datasets. The results indicate that there is no significant difference in performance among different batch sizes. This observation suggests that the generation of pseudo-labels primarily requires an approximation of the distribution information, rather than the precise accuracy of every individual label value.

#### A.4.2 Different loss functions for sequence difference measurement.

Tables 10, 11, and 12 present the comprehensive results of using different loss functions on IMDB-WIKI-DIR, AgeDB-DIR, and ECG-K-DIR, respectively. It is evident that existing methods, when augmented with Dist Loss, demonstrate superior performance on samples within few-shot regions.

#### A.4.3 Performance of Dist Loss across different imbalanced ratios

We validated the effectiveness of Dist Loss by varying the imbalance ratios of the ECG-K-DIR dataset. The data distribution diagrams are shown in Figure 5, and the corresponding results in the few-shot regions are presented in Table 13. Across eight datasets with different imbalance ratios, our method achieved the best performance in six cases and the second-best performance in the remaining two. These results collectively demonstrate the robustness of our approach across varying levels of data imbalance.

### A.5 Performance of Dist Loss on the GM Metric

We observed that on the IMDB-WIKI-DIR dataset, the performance of Dist Loss in the few-shot region, as measured by the GM metric, is inferior to that of Balanced MSE. To provide a more intuitive analysis of this phenomenon, we plotted the **sorted error distribution curves** for both Dist Loss and Balanced MSE in the few-shot region, as shown in Figure 6. Specifically, for each method, the error values were first sorted in ascending order. The x-axis represents the rank of these sorted errors, while the y-axis denotes the corresponding error magnitudes. This visualization facilitates a direct comparison of the error distributions between the two methods.

From the plot, it is evident that Dist Loss generally exhibits superior performance compared to Balanced MSE, as indicated by its curve lying below or aligning with the curve for Balanced MSE. However, a localized discrepancy is observed around the x-axis values of approximately 5 and 30,

Table 10: An ablation study on loss functions on the IMDB-WIKI-DIR dataset. $L_1$ represents MAE Loss, $L_2$ represents MSE Loss, $INV-$ denotes the probability-based inversely weighted version of these loss functions. Results on the few-shot region are reported.

| | MAE | | | | GM | | | |
|---|---|---|---|---|---|---|---|---|
| | All | Many | Median | Few | All | Many | Median | Few |
| Vanilla | 8.143 | 7.260 | 15.758 | 26.930 | 4.642 | 4.211 | 11.522 | 21.254 |
| + Dist Loss ($INV-L_1$) | 7.807 | 7.210 | 12.608 | 23.334 | 4.458 | 4.189 | 7.717 | 15.437 |
| + Dist Loss ($INV-L_2$) | 8.028 | 7.461 | 12.614 | 22.516 | 4.593 | 4.335 | 7.686 | 13.752 |
| + LDS | 8.036 | 7.445 | 12.869 | 22.753 | 4.570 | 4.322 | 7.528 | 12.803 |
| + Dist Loss ($INV-L_1$) | 8.054 | 7.545 | 12.030 | 22.178 | 4.678 | 4.486 | 6.717 | 11.334 |
| + Dist Loss ($INV-L_2$) | 8.017 | 7.479 | 12.304 | 22.331 | 4.593 | 4.369 | 7.078 | 13.021 |
| + FDS | 7.954 | 7.272 | 13.523 | 24.908 | 4.499 | 4.192 | 8.633 | 14.361 |
| + Dist Loss ($INV-L_1$) | 7.986 | 7.413 | 12.486 | 23.692 | 4.530 | 4.315 | 6.793 | 14.399 |
| + Dist Loss ($INV-L_2$) | 8.712 | 8.163 | 12.979 | 24.112 | 5.222 | 4.995 | 7.575 | 14.929 |
| + Ranksim | 7.764 | 6.956 | 14.606 | 25.999 | 4.371 | 3.996 | 9.964 | 19.690 |
| + Dist Loss ($INV-L_1$) | 7.501 | 6.888 | 12.372 | 23.894 | 4.150 | 3.910 | 7.035 | 16.036 |
| + Dist Loss ($INV-L_2$) | 7.721 | 7.129 | 12.401 | 23.772 | 4.422 | 4.183 | 7.091 | 15.422 |
| + ConR | 7.842 | 7.033 | 14.772 | 25.408 | 4.329 | 3.951 | 10.25 | 17.022 |
| + Dist Loss ($INV-L_1$) | 7.538 | 6.924 | 12.499 | 23.281 | 4.169 | 3.893 | 7.643 | 15.586 |
| + Dist Loss ($INV-L_2$) | 7.957 | 7.355 | 12.906 | 22.700 | 4.529 | 4.244 | 8.131 | 14.713 |
| + Balanced MSE | 8.033 | 7.441 | 12.768 | 23.542 | 4.716 | 4.450 | 8.035 | 12.603 |
| + Dist Loss ($INV-L_1$) | 7.788 | 7.175 | 12.732 | 23.539 | 4.460 | 4.182 | 7.900 | 15.000 |
| + Dist Loss ($INV-L_2$) | 8.075 | 7.511 | 12.625 | 22.597 | 4.616 | 4.354 | 7.754 | 14.238 |

Table 11: An ablation study on loss functions on the AgeDB-DIR dataset. $L_1$ represents MAE Loss, $L_2$ represents MSE Loss, $INV-$ denotes the probability-based inversely weighted version of these loss functions. Results on the few-shot region are reported.

| | MAE | | | | GM | | | |
|---|---|---|---|---|---|---|---|---|
| | All | Many | Median | Few | All | Many | Median | Few |
| Vanilla | 7.506 | 6.558 | 8.794 | 12.894 | 4.798 | 4.176 | 5.957 | 9.789 |
| + Dist Loss ($INV-L_1$) | 7.552 | 7.282 | 7.660 | 9.802 | 4.700 | 4.528 | 4.800 | 6.298 |
| + Dist Loss ($INV-L_2$) | 7.637 | 7.574 | 7.315 | 9.122 | 4.756 | 4.745 | 4.563 | 5.453 |
| + LDS | 7.783 | 7.070 | 8.957 | 11.279 | 5.088 | 4.599 | 6.142 | 7.846 |
| + Dist Loss ($INV-L_1$) | 7.885 | 7.635 | 8.020 | 9.872 | 5.082 | 4.964 | 5.151 | 6.109 |
| + Dist Loss ($INV-L_2$) | 7.810 | 7.341 | 8.464 | 10.437 | 5.043 | 4.752 | 5.474 | 7.051 |
| + FDS | 7.818 | 7.103 | 9.051 | 11.161 | 4.961 | 4.487 | 6.064 | 7.361 |
| + Dist Loss ($INV-L_1$) | 7.911 | 7.665 | 8.010 | 9.969 | 5.010 | 4.933 | 4.941 | 6.026 |
| + Dist Loss ($INV-L_2$) | 7.799 | 7.351 | 8.374 | 10.444 | 4.863 | 4.615 | 5.181 | 6.696 |
| + Ranksim | 7.272 | 6.363 | 8.458 | 12.569 | 4.617 | 3.939 | 6.120 | 9.495 |
| + Dist Loss ($INV-L_1$) | 7.239 | 6.605 | 7.727 | 11.877 | 4.635 | 4.194 | 5.311 | 8.164 |
| + Dist Loss ($INV-L_2$) | 7.234 | 6.506 | 7.960 | 12.102 | 4.629 | 4.097 | 5.637 | 8.515 |
| + ConR | 7.322 | 6.429 | 8.456 | 12.623 | 4.646 | 4.052 | 5.890 | 8.787 |
| + Dist Loss ($INV-L_1$) | 7.398 | 6.683 | 8.194 | 11.948 | 4.709 | 4.208 | 5.560 | 8.605 |
| + Dist Loss ($INV-L_2$) | 7.383 | 6.572 | 8.373 | 12.303 | 4.657 | 4.112 | 5.591 | 9.123 |
| + Balanced MSE | 7.663 | 7.540 | 7.353 | 9.613 | 4.658 | 4.558 | 4.511 | 6.248 |
| + Dist Loss ($INV-L_1$) | 7.537 | 7.300 | 7.540 | 9.762 | 4.751 | 4.623 | 4.737 | 6.198 |
| + Dist Loss ($INV-L_2$) | 7.633 | 7.578 | 7.288 | 9.110 | 4.718 | 4.698 | 4.505 | 5.585 |

Table 12: An ablation study on loss functions on the ECG-K-DIR dataset. $L_1$ represents MAE Loss, $L_2$ represents MSE Loss, $INV-$ denotes the probability-based inversely weighted version of these loss functions. Results on the few-shot region are reported.

| | MAE | | | | GM | | | |
|---|---|---|---|---|---|---|---|---|
| | All | Many | Median | Few | All | Many | Median | Few |
| Vanilla | 1.235 | 0.274 | 0.685 | 1.771 | 0.835 | 0.193 | 0.622 | 1.578 |
| + Dist Loss ($INV-L_1$) | 1.088 | 0.458 | 0.648 | 1.467 | 0.680 | 0.300 | 0.463 | 1.044 |
| + Dist Loss ($INV-L_2$) | 1.044 | 0.606 | 0.674 | 1.329 | 0.692 | 0.403 | 0.445 | 0.978 |
| + LDS | 1.092 | 0.368 | 0.638 | 1.510 | 0.708 | 0.236 | 0.500 | 1.190 |
| + Dist Loss ($INV-L_1$) | 1.059 | 0.463 | 0.655 | 1.413 | 0.647 | 0.291 | 0.445 | 0.984 |
| + Dist Loss ($INV-L_2$) | 1.031 | 0.557 | 0.671 | 1.325 | 0.671 | 0.363 | 0.455 | 0.957 |
| + FDS | 1.223 | 0.317 | 0.681 | 1.737 | 0.828 | 0.201 | 0.588 | 1.529 |
| + Dist Loss ($INV-L_1$) | 1.133 | 0.497 | 0.692 | 1.515 | 0.725 | 0.324 | 0.477 | 1.122 |
| + Dist Loss ($INV-L_2$) | 1.095 | 0.557 | 0.688 | 1.428 | 0.744 | 0.375 | 0.490 | 1.099 |
| + Ranksim | 1.249 | 0.275 | 0.696 | 1.791 | 0.818 | 0.215 | 0.566 | 1.510 |
| + Dist Loss ($INV-L_1$) | 1.139 | 0.394 | 0.649 | 1.577 | 0.712 | 0.317 | 0.472 | 1.099 |
| + Dist Loss ($INV-L_2$) | 1.040 | 0.587 | 0.683 | 1.325 | 0.723 | 0.400 | 0.479 | 1.031 |
| + ConR | 1.227 | 0.277 | 0.69 | 1.756 | 0.841 | 0.190 | 0.629 | 1.600 |
| + Dist Loss ($INV-L_1$) | 1.085 | 0.484 | 0.651 | 1.452 | 0.780 | 0.272 | 0.503 | 1.330 |
| + Dist Loss ($INV-L_2$) | 1.045 | 0.581 | 0.684 | 1.336 | 0.692 | 0.381 | 0.486 | 0.970 |
| + Balanced MSE | 1.106 | 0.606 | 0.727 | 1.417 | 0.824 | 0.189 | 0.620 | 1.556 |
| + Dist Loss ($INV-L_1$) | 1.092 | 0.457 | 0.65 | 1.474 | 0.678 | 0.307 | 0.443 | 1.044 |
| + Dist Loss ($INV-L_2$) | 1.046 | 0.553 | 0.658 | 1.357 | 0.696 | 0.376 | 0.480 | 0.987 |

Table 13: Performance of Dist Loss in the few-shot regions across eight datasets derived from the ECG-K-DIR dataset with varying imbalance ratios, with the best results highlighted in bold.

| | MAE | | | | | | | |
|---|---|---|---|---|---|---|---|---|
| Methods | Dataset 0 | Dataset 1 | Dataset 2 | Dataset 3 | Dataset 4 | Dataset 5 | Dataset 6 | Dataset 7 |
| Vanilla | 2.701 | 2.676 | 2.658 | 1.979 | 2.679 | 2.647 | 2.624 | 1.888 |
| LDS | 2.684 | 2.703 | 2.642 | 1.962 | 2.672 | 2.507 | 2.644 | 1.901 |
| FDS | **1.865** | 2.368 | 2.191 | **1.790** | 2.223 | 2.625 | 1.908 | 1.665 |
| Ranksim | 2.470 | 2.327 | 2.273 | 1.831 | 2.314 | 2.192 | 2.258 | 1.725 |
| ConR | 2.461 | 2.343 | 2.308 | 1.828 | 2.193 | 2.274 | 2.255 | 1.742 |
| Balanced MSE | 1.997 | 1.984 | 1.981 | 1.831 | 1.906 | 1.863 | 1.815 | 1.708 |
| Dist Loss | 1.955 | **1.873** | **1.963** | 1.822 | **1.852** | **1.803** | **1.730** | **1.638** |

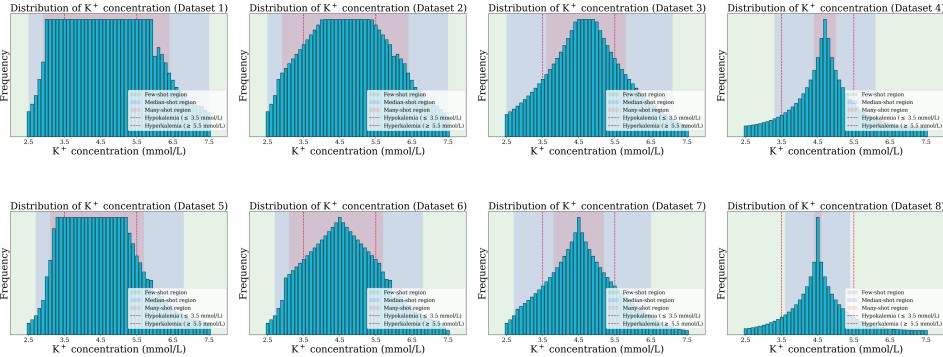

Figure 5: Data distribution diagrams for the eight datasets derived from the ECG-K-DIR dataset with varying imbalance ratios.

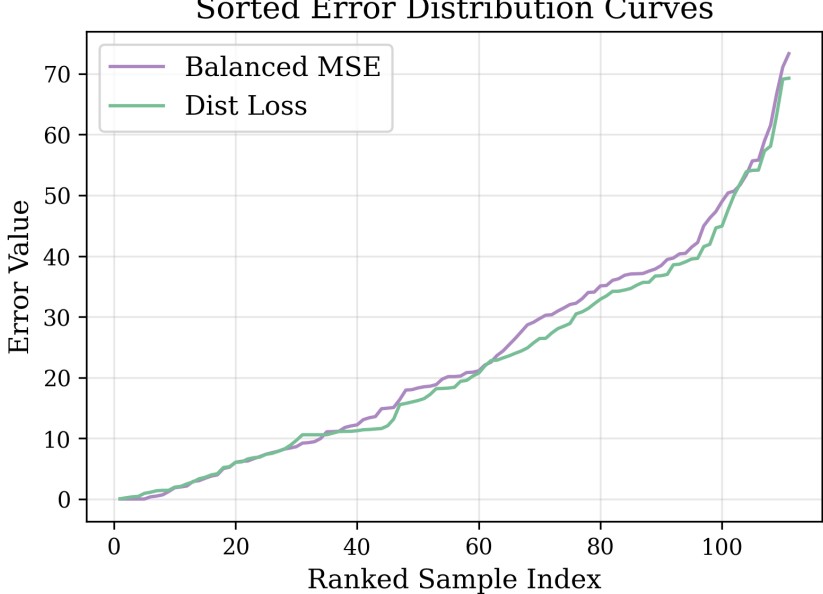

Figure 6: Sorted error distribution curves for Dist Loss and Balanced MSE in the few-shot region on the IMDB-WIKI-DIR dataset.

where the errors of Dist Loss slightly exceed those of Balanced MSE. We hypothesize that this localized discrepancy may contribute to the overall inferior performance of Dist Loss in terms of the GM metric, owing to the **cumulative multiplicative effect** intrinsic to its calculation. Unlike MAE, which averages error values, the GM metric calculates the geometric mean by multiplying error values together. This process significantly amplifies the impact of small but frequent errors. For example, consider two error distributions: $(40, 10.1, 10.1, 10.1, 10.1, 10.1)$ and $(42, 10, 10, 10, 10, 10)$. While the former achieves a lower MAE than the latter, its GM metric value is higher due to the cumulative effect, as $40 \times 1.01^5 > 42 \times 10^5$. This example underscores how the GM metric can magnify the influence of small deviations when they occur frequently.

In conclusion, the sorted error distribution curves demonstrate that Dist Loss consistently achieves better or comparable performance relative to Balanced MSE, except for minor localized discrepancies. These results suggest that the unique characteristics of the GM metric are the primary factors contributing to the observed differences in performance between the two methods.

### A.5.1 PERFORMANCE VISUALIZATION OF DIST LOSS

Figure 7 illustrates the performance comparison of three methods—vanilla model, LDS, and Dist Loss—on the IMDB-WIKI-DIR, AgeDB-DIR, and ECG-K-DIR datasets. As observed, Dist Loss achieves predictions that are systematically closer to the diagonal line ($y = x$, where the predicted values align with the ground truth values) across all datasets, indicating improved accuracy. The task on the ECG-K-DIR dataset, which involves estimating blood potassium concentration from single-lead ECG signals, is particularly challenging due to the inherently limited information provided by single-lead ECGs. This limitation exacerbates the regression dilution phenomenon, leading to larger deviations from the diagonal across all methods. Despite this difficulty, Dist Loss demonstrates superior performance, underscoring its robustness and effectiveness in addressing regression tasks with imbalanced and noisy data.

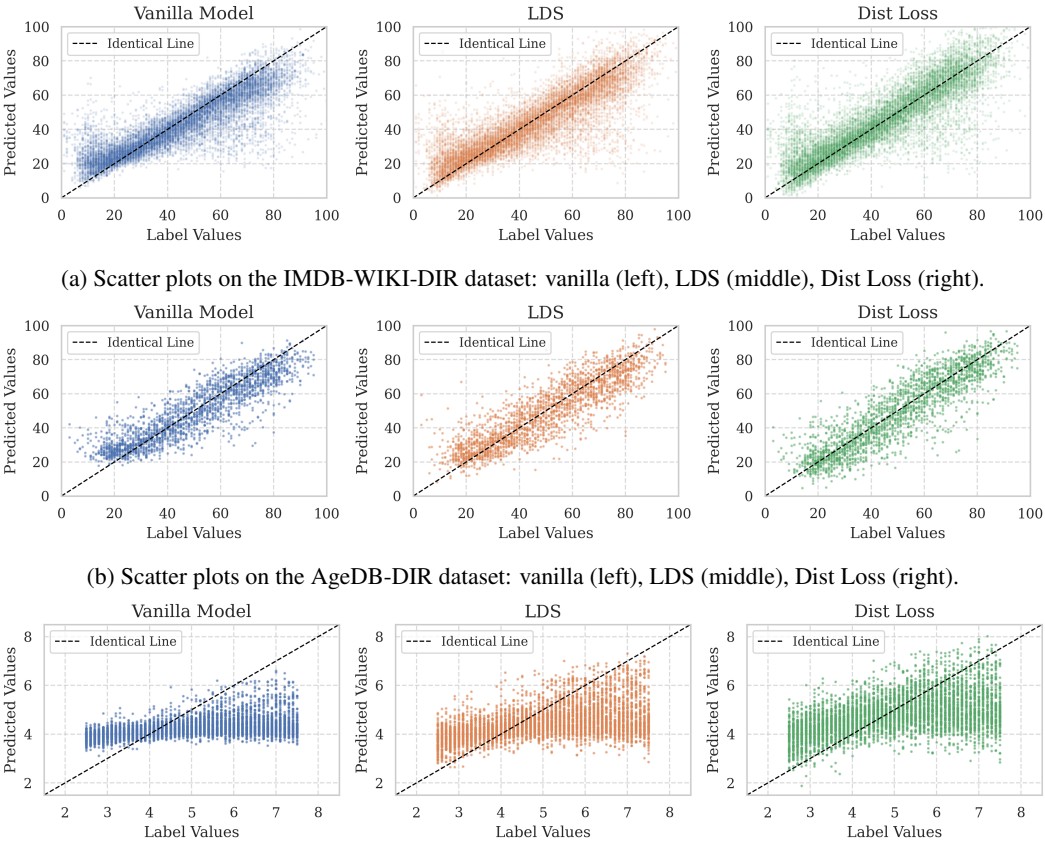

(a) Scatter plots on the IMDB-WIKI-DIR dataset: vanilla (left), LDS (middle), Dist Loss (right).

(b) Scatter plots on the AgeDB-DIR dataset: vanilla (left), LDS (middle), Dist Loss (right).

(c) Scatter plots on the ECG-K-DIR dataset: vanilla (left), LDS (middle), Dist Loss (right).

Figure 7: Performance visualization of vanilla, LDS, and Dist Loss

