# OpenReview forum: "Dist Loss: Enhancing Regression in Few-Shot Region through Distribution Distance Constraint"
_ICLR.cc/2025/Conference — ICLR 2025 Poster_

### Official Review · Reviewer_ZuRS · 2024-11-01

**Soundness:** 3
**Presentation:** 3
**Contribution:** 3
**Rating:** 8
**Confidence:** 5

**Summary:**

The paper proposes a new approach, called Dist Loss, for imbalanced deep regression. Specifically, the method generates pseudo-labels taking into consideration label distribution information, and at the same time generates pseudo-predictions. The paper is novel and well-written, but has a number of issues and can be strengthened.

**Strengths:**

The paper proposes an interesting and concise idea by generating pseudo-labels and pseudo-predictions for imbalanced deep regression. The paper is clearly written with elucidatory illustrations for easier understanding.

**Weaknesses:**

All datasets are image datasets and all models are ResNet-like models. The method has not been evaluated on more datasets like text data or tabular data.

The experiment setting may not be representative enough. For all three real-world datasets, their imbalance rates are fixed. However, the input data could be manually tweaked to generate datasets covering a wider range of imbalance rates, and to investigate the algorithm's performance in scenarios from mildly to extremely heavy-tailed distributions.

There is no theoretical analysis of the algorithm.

**Questions:**

In Figure 2, what is the exact form of the loss function L(.)? This is not clear from both the figure and the main text. Specifically, in Figure 2, for the two sequences [1, 3, 3, 4, 4, 4, 6] and [1, 2, 2, 3, 5, 6, 7], how is the loss computed?

The first step of the algorithm is discretization which maps the continuous label space Y into B bins. In such a way, the regression problem is transformed into a classification problem. Could you also show classification metrics such as overall accuracy and few-shot accuracy, along with the regression metrics, so that we have a more thorough understanding of the result? Again, what are the correlation metrics (Pearson's or Spearman's) of the results? Could you please also present some actual-by-predicted correlation plots of the prediction?

How did you determine the number of bins B?

**Details Of Ethics Concerns:**

No ethical issues.

---

> ### Author Response · Authors · 2024-11-20
>
> We sincerely appreciate your thoughtful review and insightful suggestions! Below, we have provided a comprehensive response to your questions and comments. If any of our responses fail to sufficiently address your concerns, please inform us, and we will promptly follow up.
>
> ## Diversity of datasets and models
> We sincerely appreciate your valuable feedback. The diversity of datasets and models is indeed crucial for validating the effectiveness of our study.
>
> We would like to clarify that the datasets used in this study include both image datasets (IMDB-WIKI-DIR and AgeDB-DIR) and a time-series dataset (ECG-Ka-DIR), rather than relying solely on image data. Regarding the model architecture, we adopted ResNet-50 for the image datasets, following prior related work. For the time-series dataset ECG-Ka-DIR, we employed a ResNet-like architecture due to its widespread use in ECG analysis. It is important to emphasize that Dist Loss is a **model-agnostic** approach, meaning its performance is unlikely to vary significantly with changes in the model architecture.
>
> To further enhance the diversity of datasets and models, we constructed a tabular dataset, **ECG-Ka-Tab-DIR**, based on the ECG-Ka-DIR dataset. Each column in this tabular dataset represents handcrafted features extracted from the original ECG signals (e.g., heart rate). We conducted experiments on this dataset using a two-layer multilayer perceptron (MLP) as the model. The results, presented in the table below, demonstrate that Dist Loss achieves superior performance, excelling in both overall metrics and few-shot scenarios. (Unfortunately, due to technical challenges, we were unable to provide results for FDS on this dataset. We acknowledge this limitation and appreciate your understanding.)
>
> **Table 1**: Performance comparison on the ECG-Ka-Tab-DIR dataset
>
> | Method          | MAE (Overall) | MAE (Few-shot) | GM (Overall) | GM (Few-shot) |
> |-----------------|---------------|----------------|--------------|---------------|
> | Vanilla         | 2.267         | 3.428          | 1.531        | 1.602         |
> | LDS             | 1.998         | 3.232          | 1.300        | 1.488         |
> | Ranksim         | 1.342         | 2.674          | **0.868**        | 1.839         |
> | ConR            | 1.340         | 2.670          | 0.916        | 1.836         |
> | Balanced MSE    | 1.256         | 2.443          | 0.905        | 1.597         |
> | **Dist Loss**       | **1.252**         | **2.358**          | 0.874        | **1.486**         |
>
> ## Performance under different imbalance ratios
> We greatly appreciate and thank you for your insightful comments. To further validate the effectiveness of Dist Loss, we evaluated its performance on the ECG-Ka-DIR dataset under varying imbalance ratios. The results, presented in the table below, demonstrate that our method achieved the best performance in six out of eight datasets (denoted as $D_i$, where $i$ represents the dataset index) and ranked second in the remaining two cases. These findings highlight the robustness of Dist Loss across different levels of data imbalance. For additional details, the corresponding data distribution diagrams are included in Figure 4 of the appendix in the updated version of the paper.
>
> **Tabel 2**: MAE (Few-shot) performance across different datasets $D_1$ to $D_8$ with varying imbalance ratios
>
> | Method         | $D_1$ | $D_2$ | $D_3$ | $D_4$ | $D_5$ | $D_6$ | $D_7$ | $D_8$ |
> |----------------|---------|---------|---------|---------|---------|---------|---------|---------|
> | Vanilla    | 2.701   | 2.676   | 2.658   | 1.979   | 2.679   | 2.647   | 2.624   | 1.888   |
> | LDS        | 2.684   | 2.703   | 2.642   | 1.962   | 2.672   | 2.507   | 2.644   | 1.901   |
> | FDS        | **1.865**  | 2.368   | 2.191   | **1.790**   | 2.223   | 2.625   | 1.908   | 1.665   |
> | Ranksim    | 2.470   | 2.327   | 2.273   | 1.831   | 2.314   | 2.192   | 2.258   | 1.725   |
> | ConR       | 2.461   | 2.343   | 2.308   | 1.828   | 2.193   | 2.274   | 2.255   | 1.742   |
> | Balanced MSE | 1.997 | 1.984   | 1.981   | 1.831   | 1.906   | 1.863   | 1.815   | 1.708   |
> | **Dist Loss**  | 1.955   | **1.873**   | **1.963**   | 1.822   | **1.852**   | **1.803**   | **1.730**   | **1.638**   |

---

> ### Author Response · Authors · 2024-11-20
>
> We sincerely appreciate your thoughtful feedback, which has allowed us to clarify important aspects of our study. Below, we address the questions in detail:
>
> ## The form of the loss function $L(.)$
> The form of $L(.)$ can be any loss function designed to measure the difference between two sequences based on the application requirements. In this study, we adopt an inverse probability-weighted (INV) L2 loss function as $L(.)$. Assuming the occurrence probability of a sample is $P_i$, the loss weight for that sample is $\frac{1}{P_i}$. The formula for the inverse probability-weighted L2 loss is as follows:
> $$
> \text{INV-L2 Loss} = \frac{1}{N} \sum_{i=1}^{N} \frac{(y_i - \hat{y}_i)^2}{P_i}.
> $$
> It is important to emphasize that previous studies, such as [a] and [b], have shown that solely relying on inverse probability-weighted loss functions is insufficient to address imbalanced regression problems. This limitation is why we did not include this approach as a baseline in our comparison (as the methods compared in this paper show better results) and underscores the vital role of distributional information in Dist Loss. By incorporating this information, Dist Loss more effectively captures and mitigates the discrepancies between the prediction and label distributions, thereby discouraging the model from over-predicting few-shot samples into the many-shot region. To illustrate the computation process of the loss between pseudo-labels and pseudo-predictions, consider the following example: Given pseudo-labels [1, 3, 6] and pseudo-predictions [1, 2, 7], the loss is computed in a standard manner: for each pair of elements (e.g., (1, 1), (3, 2), (6, 7)) at corresponding indices, the loss is calculated, and then the average of these values is taken.
>
> ## Discretization of the label space and determination of the number of bins
> The discretization of the label space in this paper serves to facilitate the use of label distribution information without altering the essence of the regression problem (e.g., transforming it into a classification problem). In practical computation, continuous probability density curves are challenging to process directly; therefore, they are discretized into bins to estimate and utilize the probability distribution.
>
> The number of bins depends on the resolution of the labels relevant to the specific problem. For example, the resolution for age could be 1 year, for blood potassium concentration it might be 0.1 mmol/L, and for temperature, it could be 0.1°C. Once the resolution is determined, the number of bins can be calculated accordingly. For instance, with a bin width of 1 for age, the smallest possible value is 0 and the largest is 150, resulting in 151 bins. In practice, some label values may not occur (e.g., 150 years of age), but this does not impact the task, as we are merely defining a sufficiently large range to accommodate all possible labels.
>
> We hope this explanation clarifies your concerns and thank you again for providing the opportunity to elaborate on these points. Your feedback is invaluable for refining our work.
>
> ## References
> [a] Yang, Yuzhe, et al. "Delving into deep imbalanced regression." International conference on machine learning. PMLR, 2021.
>
> [b] Ren, Jiawei, et al. "Balanced mse for imbalanced visual regression." Proceedings of the IEEE/CVF Conference on Computer Vision and Pattern Recognition. 2022.

---

> > ### Author Response · Authors · 2024-11-20
> >
> > ## Additional evaluation results based on classification metrics and Pearson correlation coefficient
> > We appreciate the opportunity to clarify this point. The regression task can indeed be evaluated using classification metrics by transforming it into a three-class classification problem. In this setup, the target space is divided into many-shot, median-shot, and few-shot regions. The model’s predicted values are compared with the corresponding labels to determine whether they fall into the same region. The performance of various methods in both overall and few-shot regions is summarized in the table below, evaluated using accuracy and F-score as metrics. Additionally, we report the Pearson correlation coefficient between the predicted values and the labels within each region. It is worth noting that overall accuracy and correlation coefficients are sensitive to data imbalance, as the few-shot region contains fewer samples and thus has a limited contribution to these overall metrics. Overall, the results demonstrate that Dist Loss achieves outstanding performance in the few-shot region, closely aligning with the motivation discussed in this paper, which emphasizes the importance of rare yet highly informative few-shot samples.
> >
> > **Table 3**: Performance comparison of different methods on ECG-Ka-DIR (ECG-Ka), AgeDB-DIR (AgeDB), and IMDB-WIKI-DIR (IMDB): accuracy (Acc) and Pearson correlation coefficient (Corr) for overall (O) and few-shot (F) scenarios
> > | Method         | ECG-Ka Acc O | ECG-Ka Acc F | ECG-Ka Corr O | ECG-Ka Corr F | AgeDB Acc O | AgeDB Acc F | AgeDB Corr O | AgeDB Corr F | IMDB Acc O | IMDB Acc F | IMDB Corr O | IMDB Corr F |
> > |----------------|--------------|--------------|---------------|---------------|-------------|-------------|--------------|--------------|------------|------------|-------------|-------------|
> > | Vanilla        | 0.252        | 0.055        | 0.457         | 0.442         | 0.754       | 0.093       | 0.887        | **0.957**        | 0.921      | 0.027      | 0.851       | 0.767       |
> > | LDS            | 0.332        | 0.197        | 0.489         | 0.481         | 0.767       | 0.216       | 0.877        | 0.943        | 0.916      | 0.162      | 0.845       | 0.769       |
> > | FDS            | 0.26         | 0.179        | 0.438         | 0.433         | **0.776**   | 0.247       | 0.875        | 0.946        | 0.915      | 0.126      | 0.845       | 0.735       |
> > | Ranksim        | 0.24         | 0.04         | 0.439         | 0.422         | 0.768       | 0.099       | **0.892**    | 0.947        | 0.922      | 0.081      | **0.858**   | 0.766       |
> > | ConR           | 0.256        | 0.066        | 0.459         | 0.444         | 0.770       | 0.142       | 0.890        | 0.953        | **0.923**  | 0.090      | 0.855       | 0.760       |
> > | Balanced MSE   | 0.354        | 0.294        | 0.451         | 0.439         | 0.732       | **0.358**       | 0.887        | **0.957**        | 0.894      | 0.153      | 0.852       | 0.770       |
> > | Dist Loss      | **0.377**    | **0.332**    | **0.487**     | **0.501**     | 0.737       | 0.34    | 0.887        | **0.957**        | 0.894      | **0.207**  | 0.849       | **0.791**   |

---

> > > ### Comment · Reviewer_ZuRS · 2024-11-26
> > > **Thank you for the reply!**
> > >
> > > Hi Authors, thanks for the detailed reply and the new results. While I'll modify my initial rating accordingly, I still think it could be better to include some actual-by-predicted correlation plots to visually show the goodness of the prediction.

---

> > > > ### Author Response · Authors · 2024-11-27
> > > >
> > > > Dear Reviewer ZuRS,
> > > >
> > > > Thank you for your thoughtful and constructive feedback, as well as for revisiting your initial rating. We greatly appreciate your suggestion to include actual-by-predicted correlation plots to enhance the visualization of prediction quality. In response, we have incorporated scatter plots in the updated PDF (see Figure 6), which clearly illustrate the relationship between predicted and ground truth values. We hope this addition provides the clarity you were seeking and further demonstrates the effectiveness of our method.
> > > >
> > > > We are deeply grateful for your valuable input, which has significantly contributed to improving the quality and presentation of our work.
> > > >
> > > > Best regards,
> > > >
> > > > Submission #4532 Authors

---

### Official Review · Reviewer_7CfX · 2024-11-03

**Soundness:** 2
**Presentation:** 1
**Contribution:** 2
**Rating:** 6
**Confidence:** 3

**Summary:**

The paper addresses the challenge of imbalanced data distributions in regression tasks, especially the difficulty deep learning models face in accurately predicting few-shot settings. The authors introduce a new loss function, called Dist Loss, designed to align the distribution of model predictions with the label distribution using kernel density estimation and pseudo-predictions. The authors also contribute a differentiable relaxation for their loss function to make the model end-to-end training. The paper conducts experiments on the regression of three datasets, compares with five related works, and has demonstrated significant improvements.

**Strengths:**

The paper has the following strengths:

(i) The problem of imbalanced regression tasks is important to study but less well investigated in related works. The authors contributed a new loss function to train the model and showed good results compared to other baselines in two aspects: using the new loss function integrated into the baseline and combining the existing training loss with their proposed one. The running time is as fast as the mean squared error and significantly faster than other baselines.

(ii) The idea of using a pseudo-label to define the distance between prediction and label distribution is interesting. Also, the authors applied different sorting techniques to make their Dist Loss ($L(S_P, S_L)$) trainable.

**Weaknesses:**

While the paper achieved good results, its presentation lacks details and information to help the reviewer understand core ideas and how the method efficiently addresses imbalanced problems in regression tasks.

(i) In Figure 2, what is the role of **"Labels"** and **"Predictions"** terms (inside **Dist loss calculation**) while the Loss function already received **Pseudo-labels** and **Sorted Prediction**?

(ii) In Section 3.2.1, authors jumped into detail about how they compute pseudo-labels and pseudo-predictions but they have not addressed two key questions:
+ Why do we need to compute "pseudo-here"? How is it important to deal with the imbalanced dataset?
+ Section 3.2.1 is based on the Empirical label distribution (KDE), but no details about KDE are discussed (either background and how it is used), which makes the readers unable to follow the details later. For e.g., is the KDE step applied at a batch size level or at the whole training dataset?

Another unclear problem is the standard regression formulation, which transforms continuous label space Y into discretized B bins (**Section 3.1**). How is the standard learning scheme trained with these discretized labels?

(iii) The descriptions in Equation (1) and (2) and the definition of $N_{L'}$ are also hard to imagine without a figure to illustrate core ideas beyond.

Overall, the paper's methodology section is not well presented, with information presented sporadically, and does not address core concerns about imbalanced settings.

**Questions:**

The reviewer has the following questions:


(a) can you explain how the KDE is used in your formulation? What is input data? Do we need to use it in the testing part or only in the training steps?

(b) can you explain why the pseudo-label is important to overcome imbalances in regression?

(c) The ablation study in Table 4 is difficult to capture. In particular, the authors mentioned, *"We demonstrate the effects of using different functions, considering the probability-based inversely weighed MAE and MSE losses."* There are two questions here. First, what is *probability-based inversely* (INV) ? and why $INV - L_1$ or $ INV - L_2$ shows benefit of your proposed one?

(d) because dist_loss works with discrete values in sorted orders, how can you make the training process stable and avoid exploding gradients given large values from the loss function?

---

> ### Author Response · Authors · 2024-11-20
>
> We sincerely appreciate your thoughtful review and insightful suggestions! Below, we have provided a comprehensive response to your questions and comments. If any of our responses fail to sufficiently address your concerns, please inform us, and we will promptly follow up.
>
> ## The core idea of Dist Loss and its utility for imbalanced regression
> We sincerely apologize for any confusion and appreciate the opportunity to clarify the core concept behind Dist Loss. As illustrated in Figure 1, a primary challenge in imbalanced regression is the mismatch between the predicted value distribution (prediction distribution) and the true label distribution. This discrepancy arises because the model, when minimizing overall error, tends to over-predict samples from the few-shot region into the many-shot region. While this reduces overall error, it severely impacts prediction accuracy in the few-shot region. To address this, we propose Dist Loss, which explicitly minimizes the distance between the prediction distribution and the label distribution during optimization. This approach **discourages the model from over-predicting few-shot samples into the many-shot region**, thus improving accuracy in the few-shot region. The implementation of Dist Loss introduces an additional loss term that quantifies the distance between the label distribution and the prediction distribution, complementing the standard model optimization. This involves constructing two key components: pseudo-labels and pseudo-predictions, which represent the distributional information of the true labels and predicted values, respectively. The distance between these components is calculated using a loss function $L(\cdot)$, which approximates the difference between the label and prediction distributions. Additionally, the same loss function $L(\cdot)$ is used in the standard optimization process to compute the difference between individual predictions and their corresponding true labels. By combining these terms, Dist Loss aligns the prediction distribution with the label distribution, reducing errors in the few-shot region while maintaining strong overall performance.
>
> We hope this explanation clarifies the motivation and mechanism behind Dist Loss. Please let us know if further details are required.

---

> ### Author Response · Authors · 2024-11-20
>
> ## How KDE is applied
> We apologize for any confusion and appreciate your valuable feedback. Below, we provide a detailed explanation of how Kernel Density Estimation (KDE) is applied in our method:
>
> **1. Purpose of KDE**: KDE is a non-parametric method used to estimate the probability density function of a random variable without assuming a specific distribution form. In our method, we apply KDE to the labels in the entire training dataset to estimate the probability density of the label distributions.
>
> **2. Input and Output**: The input to KDE is the set of all labels in the training dataset, and the output is a continuous probability density function of the labels. For simplification, we discretize this continuous density function into $B$ bins (intervals), transforming it into a discrete probability distribution. This discretization is employed solely to facilitate the utilization of label distribution information and does not impact the normal training process.
>
> **3. Pseudo-label Generation at the Batch Level**: For each training batch, we calculate the expected occurrence count $n_i$ of each label $y_i$ based on the label distribution. This is done to align with the frequency of parameter updates in the model, using the formula: $n_i = \text{batch size} \times P(y_i)$, where $P(y_i)$ is the probability of label $y_i$ in the discretized label distribution. Since $n_i$ is usually a decimal, we round it down to the nearest integer. However, this rounding may cause the sum of $n_i$ values to deviate from the batch size. To address this, we adjust $n_i$ values using Equation (1) to ensure the adjusted values $n_i'$ sum up to the batch size. The final set of $n_i'$ values is denoted as $N_L'$.
>
> **4. Construction of Pseudo-labels**: Using the adjusted $n_i'$ values, we generate pseudo-labels. For example, if the label set is [1, 2, 3] and the adjusted counts are [1, 3, 2], we repeat label $1$ once, label $2$ three times, and label $3$ twice, resulting in the pseudo-label set [1, 2, 2, 2, 3, 3]. This process is formally described in Equation (2), which defines the mapping between pseudo-label indices and label values.
>
> **5. How Pseudo-labels Capture Label Distribution Information**: The pseudo-labels generated in this way approximate the original label distribution. If we plot a histogram of the pseudo-labels, it will closely resemble the discretized label probability distribution. This process effectively captures the label distribution information, which is then used for supervised training of the model.
>
> **6. KDE in the Testing Phase**: It is important to note that KDE is only used in the training phase for pseudo-label construction. During the testing phase, the model is evaluated directly on the test data predictions, without utilizing KDE.
>
> We hope this explanation clarifies the role and implementation of KDE in our method. If you have further questions or need additional clarifications, we are happy to address them promptly.

---

> ### Author Response · Authors · 2024-11-20
>
> ## Definition of INV- and objectives of the ablation study on loss functions $L(.)$
>
> "Probability-based inversely weighted (INV)" refers to adjusting the contribution of each sample to the loss based on the inverse of its occurrence probability. Specifically, if a sample's occurrence probability is $P_i$, its loss is weighted by $1/P_i$.
>
> For example, consider L1 Loss, which is typically defined as:
>
> $$
> \text{L1 Loss} = \frac{1}{N} \sum_{i=1}^{N} |y_i - \hat{y}_i|
> $$
>
> For INV-L1 Loss, the expression is modified as follows:
>
> $$
> \text{INV-L1 Loss} = \frac{1}{N} \sum_{i=1}^{N} \frac{|y_i - \hat{y}_i|}{P_i}
> $$
>
> As mentioned in the paper, Dist Loss measures the differences between pseudo-labels and pseudo-predictions, as well as between true labels and model predictions, using a loss function $L(.)$. This loss function $L(.)$ can be flexibly chosen based on the application requirements. In Table 4 (and the corresponding appendix), we conduct ablation studies to illustrate the effects of using different $L(.)$ functions. For example, INV-L2 generally emphasizes accuracy in the few-shot region due to its higher sensitivity to large errors, making it more effective when the focus is on improving performance in under-represented areas. On the other hand, INV-L1 achieves a balance between overall accuracy and few-shot region accuracy (with a greater emphasis on the latter), making it suitable for scenarios where both aspects are crucial. This highlights the flexibility of Dist Loss, allowing practitioners to design $L(.)$ based on specific application scenarios to achieve tailored performance improvements. However, it is important to emphasize that previous studies, such as [a] and [b], have demonstrated that solely relying on inverse probability-weighted loss functions is insufficient to effectively address imbalanced regression problems. This is why we incorporate distributional information in Dist Loss, which helps to more effectively capture and mitigate the discrepancies between the prediction and label distributions.
>
> We sincerely appreciate your insightful feedback and are happy to provide these clarifications. If you have any further questions, we are happy to address them promptly.
>
> ## References
> [a] Yang, Yuzhe, et al. "Delving into deep imbalanced regression." International conference on machine learning. PMLR, 2021.
>
> [b] Ren, Jiawei, et al. "Balanced mse for imbalanced visual regression." Proceedings of the IEEE/CVF Conference on Computer Vision and Pattern Recognition. 2022.

---

> > ### Author Response · Authors · 2024-11-20
> >
> > ## Stability assurance of Dist Loss during training
> > In our experiments, we observed that the training process with Dist Loss remained stable. This can be attributed to the following key factors:
> >
> > **1. Fine-tuning with Pre-trained Model Parameters**: During training, we employed pre-trained model parameters and fine-tuned only the linear layers. This strategy, consistent with the approach used for Balanced MSE, helps maintain stability throughout the training process and prevents issues such as exploding gradients.
> >
> > **2. Numerical Consistency Between Components of Dist Loss**: Dist Loss introduces an additional term that measures the difference between pseudo-labels and pseudo-predictions, reflecting the discrepancy between the corresponding distributions. Since pseudo-labels are sampled from the training set labels and pseudo-predictions are derived from the sorted model predictions, their values are on the same scale as the actual labels and predictions. This ensures that the magnitude of the loss term for pseudo-labels and pseudo-predictions aligns closely with the standard regression loss term for labels and predictions. This numerical consistency significantly mitigates potential instability during training and reduces the likelihood of gradient explosion.
> >
> > We sincerely appreciate your feedback, which allowed us to clarify the stability considerations of Dist Loss. Should you have any further questions or suggestions, we would be happy to address them.

---

> > > ### Comment · Reviewer_7CfX · 2024-11-25
> > > **Response to authors**
> > >
> > > Dear Authors,
> > >
> > > Thank you a lot for your responses; it made me understand your algorithm better. I increased my score from 5 to 6. However, I would emphasize that in case this paper is accepted by AC decision, please take into account the above explanations to make major improvements in the method section, either writing part or coming up with a better figure for the main method if possible.
> > >
> > > Regards

---

> > > > ### Author Response · Authors · 2024-11-25
> > > >
> > > > Dear Reviewer 7CfX,
> > > >
> > > > Thank you very much for your kind feedback and for taking the time to review our paper again. We are truly grateful for your increased score and your constructive comments, which have helped us better identify areas for improvement.
> > > >
> > > > We sincerely appreciate your suggestions regarding the method section and the main figure. We will carefully incorporate these valuable recommendations to enhance the clarity and presentation of our work, ensuring that the writing and illustrations better convey our ideas.
> > > >
> > > > Once again, thank you for your thoughtful evaluation and for contributing to the improvement of our work.
> > > >
> > > > Best regards,
> > > >
> > > > Submission #4532 Authors

---

### Official Review · Reviewer_C9ay · 2024-11-08

**Soundness:** 3
**Presentation:** 3
**Contribution:** 3
**Rating:** 6
**Confidence:** 3

**Summary:**

This paper tackles the issue of imbalanced regression, which often results in poor performance in regions with few samples, by proposing a new loss function termed Dist Loss. Dist Loss is designed to reduce the distribution gap between model predictions and actual labels, thereby promoting closer alignment with the target distribution, particularly in sparse data regions. The approach has shown enhanced performance in few-shot scenarios across several datasets, such as IMDB-WIKI-DIR, AgeDB-DIR, and ECG-Ka-DIR, and complements existing techniques well. Through extensive experimentation, Dist Loss demonstrates state-of-the-art accuracy for rare instances, highlighting its relevance for critical fields like healthcare.

**Strengths:**

1. The paper introduces Dist Loss, a new loss function specifically designed to tackle the problem of imbalanced regression, improving model performance in low-sample regions.

2. Dist Loss is easy to incorporate with current techniques and enhances their performance in few-shot areas, making it a practical addition to existing approaches.

3. The proposed method is evaluated on different datasets.

**Weaknesses:**

1. According to the problem setting, the distribution has been discretized using fixed bin size, which can also be adapted into classification setting if one have a look at Figure 1. However the authors claimed in section 2.2 that the methods developed for imbalanced image classification can not be well adapted into imbalanced regression task. From my point of view, if bin is chosen then classification based method can also be used in regression task. More explanation should be provided by the authors.


2. The performance gain of the proposed loss function is not promising compared with Balanced MSE in Table I. It even underperforms BalancedMSE on IMDB/WIKI/DIR dataset using GM metric.

3. The authors are also suggested to compare with some other existing related works, e.g., a.

In a and b, the task is evaluated in four different settings, which are all, many, medium, few, which is also mentioned by the authors. However it seems in the provided Tables there are only results for few shot regions. I find the original different shot settings more convincing since it can showcase the generalizability of the proposed loss function among different shot settings.

Another question, why the reported performance of ConR is different compared with the performance reported by their paper for AgeDB DIR dataset? Did the authors follow the same experimental setting as introduced in ConR and other existing works?

a. Wang, Z., & Wang, H. (2024). Variational imbalanced regression: Fair uncertainty quantification via probabilistic smoothing. Advances in Neural Information Processing Systems, 36.

b. Keramati, M., Meng, L., & Evans, R. D. (2023). Conr: Contrastive regularizer for deep imbalanced regression. ICLR 2024.

4. In Table 3, it seems that balanced MSE is better than the proposed Dist Loss regarding the training time consumption. The benefits from the training efficiency pespective is now well highlighted.


5. I have some concerns with Table 4. From the paper of ConR as mentioned in b, LDS with ConR can achieve better performance on AgeDB DIR dataset, with 9.62 for MAE and 6.87 for AgeDB DIR when we considering the few shot setting. The authors are suggested to compare with all other existing works in Table 4.

**Questions:**

1. In Section 2.2, the authors claim that imbalanced image classification methods cannot be well adapted for imbalanced regression. However, with the chosen discretized distribution using fixed bin sizes, wouldn't classification-based methods be applicable? Could the authors elaborate on why classification-based methods might not work well in this setting and clarify the limitations or challenges?

2. The proposed loss function does not show a strong performance improvement over Balanced MSE, as shown in Table I, and it even underperforms Balanced MSE on the IMDB/WIKI/DIR dataset using the GM metric. Can the authors explain why the proposed loss function struggles to outperform Balanced MSE in these cases and discuss any specific scenarios where it demonstrates advantages?

3.Could the authors compare their method with other existing works, such as Wang & Wang (2024) and Keramati et al. (2023)? These works evaluate the task across different settings (all, many, medium, few), which seems like a more convincing framework to assess generalizability. Why did the authors choose not to include similar shot-based evaluations? Would this additional analysis provide deeper insights into the model’s robustness?


4.The performance of ConR reported in the paper differs from what is reported in the original ConR paper for the AgeDB DIR dataset. Did the authors follow the same experimental settings as in these related works? If there were any differences in the setup, could they clarify these and explain how they might impact the results?

5.Table 3 shows that Balanced MSE is more efficient in terms of training time compared to the proposed Dist Loss. Could the authors discuss the trade-offs between training time efficiency and performance gain? Given the additional time cost, what are the practical advantages of using Dist Loss?
Performance on AgeDB DIR Dataset in Few-Shot Settings:

6. According to Keramati et al. (2023), ConR combined with LDS achieved better results on the AgeDB DIR dataset in the few-shot setting (MAE of 9.62 and 6.87). Why does the proposed method seem to underperform compared to this? Did the authors consider integrating ConR or LDS in their approach to enhance few-shot performance, and if not, what were the reasons?

---

> ### Author Response · Authors · 2024-11-20
>
> Thank you for your constructive review and valuable suggestions! Below, we provide a detailed response to your questions and comments. If any of our responses fail to sufficiently address your concerns, please inform us, and we will promptly follow up.
>
> ## Why methods for imbalanced classification are not well-suited for imbalanced regression
> We appreciate your thoughtful comments regarding the challenges of imbalanced regression tasks. While it is possible to transform regression problems into classification tasks by discretizing the regression labels, we argue that this approach introduces several limitations. Below, we outline the key reasons why methods for imbalanced classification are not directly applicable to imbalanced regression:
>
> **1. Nature of Regression vs. Classification**: Classification tasks have well-defined category boundaries, whereas regression tasks involve continuous and unbounded labels. Discretization disrupts the continuity inherent to regression tasks.
>
> **2. Semantic Meaning of Label Distances**: The distances between labels in regression tasks carry meaningful information that can facilitate better representation learning. For instance, in age prediction, samples with a label of 31 may be sparse, but the neighboring samples with labels 30 and 32 might be abundant. Consequently, the model's prediction error for 31-year-old individuals can be reduced by leveraging the information from adjacent labels. In contrast, classification tasks treat category values as arbitrary identifiers without intrinsic meaning, preventing the model from utilizing information from neighboring samples, which hinders effective learning.
>
> **3. Purpose of Discretization in Our Approach**: It is important to emphasize that in our work, discretizing the label space is intended solely to facilitate the utilization of label distribution information, without altering the fundamental nature of the regression problem (e.g., converting it into a classification problem). In practice, continuous probability density curves are challenging to process directly. Thus, dividing the label space into bins offers a practical approach for estimating and utilizing the probability distribution.
>
> ## Comparative performance analysis with Balanced MSE
> **1. Overall Performance**: We appreciate your insightful comments and the opportunity to discuss the relative performance of our method. While Balanced MSE performs well in the few-shot region, our method outperforms it overall, achieving better results in 5 out of 6 evaluation metrics across 3 datasets. For example, in terms of MAE, we observe relative improvements of 4.42%, 5.11%, and 6.21% on the IMDB-WIKI-DIR, AgeDB-DIR, and ECG-Ka-DIR datasets, respectively. Additionally, a Wilcoxon signed-rank test confirms that our method is statistically superior, with p-values of 0.01, 0.02, and 6.95e-44 for the three datasets.
>
> **2. On GM Metric for IMDB-WIKI-DIR Dataset**: We observe that Dist Loss achieves a higher GM metric than Balanced MSE on the IMDB-WIKI-DIR dataset. To explore this, we plotted the sorted error distribution curves for both methods in the few-shot region (Figure 5 of the updated paper). While Dist Loss generally outperforms Balanced MSE, small discrepancies around error ranks 5 and 30 may contribute to its slightly inferior GM performance. This is because the GM metric amplifies the effect of small frequent errors, unlike MAE, which averages them. For instance, the GM of (40, 10.1, 10.1, 10.1, 10.1, 10.1) exceeds that of (42, 10, 10, 10, 10, 10), despite the former having a lower MAE. This highlights how frequent small errors can disproportionately impact the GM metric.
>
> **3. Additional Metrics**: We expanded our analysis by incorporating additional evaluation metrics for the IMDB-WIKI-DIR dataset. Specifically, we transformed the regression problem into a three-class classification task to verify whether the model's predicted values fall within the same region (many-shot, median-shot, or few-shot) as the true labels. We categorized the predictions into these regions and evaluated performance using accuracy and F1-score for each region. Furthermore, we computed the Pearson correlation coefficient to assess the alignment of predicted values with the true labels. The results from these additional metrics indicate that our method consistently outperforms Balanced MSE, demonstrating that our predictions are more accurately aligned with the true label regions across all evaluation criteria.
>
> **Table 1**: Performance comparison for IMDB-WIKI-DIR dataset (Few-shot region)
> | Methods      | Accuracy | F1-score | Pearson Correlation Coefficient |
> |--------------|----------|----------|---------------------------------|
> | Balanced MSE | 0.153    | 0.266    | 0.770                           |
> | Dist Loss    | 0.207    | 0.343    | 0.791                           |

---

> ### Author Response · Authors · 2024-11-20
>
> ## Comparison with variational imbalanced regression (VIR)
> Thank you for your valuable suggestion regarding the comparison with VIR. We added experimental results for VIR on the IMDB-WIKI-DIR and AgeDB-DIR datasets, as used in the original VIR paper, shown in the table below. Our method outperforms VIR in the few-shot region, achieving MAE and GM values of 9.122 and 5.453 on AgeDB-DIR (VIR: 10.427, 7.237), and 22.516 and 13.752 on IMDB-WIKI-DIR (VIR: 23.483, 15.857). Additionally, combining VIR with Dist Loss yields even better results: on AgeDB-DIR, the MAE and GM are 8.159 and 5.145; on IMDB-WIKI-DIR, they are 21.908 and 13.006. This highlights the complementary advantage of Dist Loss in improving existing methods.
>
> **Table 2**: Comparison with VIR on the IMDB-WIKI-DIR and AgeDB-DIR datasets
> | Methods         | IMDB-WIKI-DIR MAE | AgeDB-DIR MAE | IMDB-WIKI-DIR GM | AgeDB-DIR GM |
> |-----------------|-------------------|---------------|------------------|--------------|
> | VIR             | 23.483            | 10.427        | 15.857           | 7.237        |
> | Dist Loss       | 22.516            | 9.122         | 13.752           | 5.453        |
> | VIR + Dist Loss | 21.908            | 8.159         | 13.006           | 5.145        |
>
> ## Why the overall/many/median/few-shot region setup was not used
> **1. Detailed Results in the Appendix**: We appreciate your insightful concern. As noted in the paper, we have included detailed results for the overall, many-shot, median-shot, and few-shot regions for each dataset in the appendix. These results demonstrate that our method not only maintains overall model performance but also significantly improves performance in the few-shot (and median-shot) regions.
>
> **2. Focus on the Few-shot Region**: As emphasized throughout the paper, our primary focus is on improving prediction accuracy in the few-shot region, as samples in this region often have more substantial real-world implications. This aligns with the core objectives of imbalanced regression research, as demonstrated in studies [a], [b], and [c]. For instance, in ECG-based potassium concentration prediction, samples corresponding to hyperkalemia and hypokalemia typically fall within the few-shot region (as illustrated in Figure 1 of the paper). Predicting these rare but critical cases accurately is essential in clinical contexts, as abnormal potassium levels can lead to life-threatening conditions such as ventricular fibrillation or cardiac arrest. In contrast,  the prediction accuracy of samples within the normal potassium range (many-shot region) is less practically significant, as it does not result in clinically meaningful consequences.
>
> ## Why the experimental results of ConR differ from the original paper
> **1. Differences in Experimental Setup**: We appreciate your observation regarding the differences in experimental results for ConR. As outlined in the Appendix of our paper, we followed the experimental settings of the original ConR paper and other related works in all respects, with the exception of the number of GPUs used and the number of training epochs. Specifically:
>
> ***1.1. Number of GPUs***: The original ConR paper utilized 4 GPUs in PyTorch's data parallel mode, while we conducted all our experiments using a single GPU.
>
> ***1.2. Number of Training Epochs***: The original ConR paper trained for 120 epochs, whereas other works, such as LDS, FDS, and RankSim, trained for 90 epochs. To maintain consistency across all methods, we standardized the number of training epochs to 90.
>
> **2. Consistency with Open-source Code**: Furthermore, we utilized the open-source code provided by previous studies (available on GitHub) for our experiments, without any further modifications to the experimental settings. We want to emphasize that no intentional alterations were made to the code that could affect the fairness of our comparisons.
>
> ## References
> [a]. Branco, P., Torgo, L., & Ribeiro, R. P. (2017, October). SMOGN: a pre-processing approach for imbalanced regression. In First international workshop on learning with imbalanced domains: Theory and applications. PMLR.
>
> [b]. Moniz, N., Ribeiro, R., Cerqueira, V., & Chawla, N. (2018, October). Smoteboost for regression: Improving the prediction of extreme values. In 2018 IEEE 5th international conference on data science and advanced analytics (DSAA). IEEE.
>
> [c]. Steininger, M., Kobs, K., Davidson, P., Krause, A., & Hotho, A. (2021). Density-based weighting for imbalanced regression. Machine Learning, 110, 2187-2211.

---

> > ### Author Response · Authors · 2024-11-20
> >
> > ## Time Complexity of Dist Loss
> > We appreciate your interest in the time complexity of Dist Loss. As shown in Table 3, both Dist Loss and Balanced MSE exhibit significantly lower time costs compared to other methods. When comparing Dist Loss to Balanced MSE, we believe their time consumption is of the same order of magnitude. The only additional operation in Dist Loss is sorting the model's predictions within each batch. As discussed in [d], this sorting operation has a time complexity of $O(n \log n)$, which is computationally efficient. In practice, the contribution of this per-batch sorting to the overall training time is minimal, as the forward and backward passes of the model dominate the time cost.
> >
> > ## Comparison with ConR + LDS
> > We appreciate your insightful comment regarding the comparison with ConR + LDS. We would like to clarify that the training strategy for ConR + LDS in the original paper differs from the approach we use in our study. Specifically, the original ConR paper trains all layers of the model, whereas our method fine-tunes only the linear layers of the model based on the pretrained parameters from ConR (similar to the strategy used in Balanced MSE). This approach was chosen to facilitate the combination with various methods, as both ConR and RankSim optimize their models primarily through their respective loss functions. Our goal is to avoid the complexity of adding an increasing number of loss functions that require careful hyperparameter tuning to balance effectively. The results in Table 4 of our paper show that our method improves the performance of existing methods, particularly in the few-shot region, by fine-tuning only the linear layers of the model. To better compare our approach with ConR + LDS, we retrained all layers of the model using ConR + Dist Loss. The experimental results are presented in the table below. As can be seen, our method outperforms ConR + LDS in the overall region and shows significant improvements in both the median and few-shot regions. These findings suggest that Dist Loss provides a substantial performance boost to methods like ConR, further supporting its effectiveness in imbalanced regression tasks.
> >
> > **Table 3**: Performance comparison: ConR + LDS vs. ConR + Dist Loss on AgeDB-DIR dataset
> >
> > | Method              | MAE (Overall) | MAE (Many-shot) | MAE (Median-shot) | MAE (Few-shot) | GM (Overall) | GM (Many-shot) | GM (Median-shot) | GM (Few-shot) |
> > |---------------------|---------------|-----------------|-------------------|----------------|--------------|----------------|------------------|---------------|
> > | **ConR + LDS**      | 7.477         | 6.828           | 8.565             | 10.59          | 4.826        | 4.427          | 5.666            | 6.991         |
> > | **ConR + Dist Loss**| 7.341         | 6.921           | 7.845             | 9.918          | 4.544        | 4.324          | 4.875            | 5.984         |
> >
> > ## References
> > [d]. Blondel, M., Teboul, O., Berthet, Q., & Djolonga, J. (2020, November). Fast differentiable sorting and ranking. In International Conference on Machine Learning. PMLR.

---

> > > ### Comment · Reviewer_C9ay · 2024-11-25
> > > **Response**
> > >
> > > Thank the authors for the detailed response, the experimental setting regarding comparison with ConR + LDS you mentioned in your rebuttal would be better to be clarified in your main paper. Apart from that I think most of my concerns are solved. I will increase my score to 6. Looking forward to your released code.

---

> > > > ### Author Response · Authors · 2024-11-25
> > > >
> > > > Dear Reviewer C9ay,
> > > >
> > > > Thank you for your thoughtful feedback and for revisiting our paper. We are glad to hear that our responses addressed most of your concerns. We appreciate your suggestion to further clarify the experimental setting regarding the comparison with ConR + LDS in the main paper, and we will make sure to incorporate this in future revisions.
> > > >
> > > > We are also committed to releasing the code upon acceptance to ensure reproducibility and to benefit the research community.
> > > >
> > > > Thank you again for your time and valuable comments!
> > > >
> > > > Best regards,
> > > >
> > > > Submission #4532 Authors

---

### Comment · Area_Chair_QczK · 2024-11-24
**discussion**

Dear reviewers,

Thank you for your contribution. Soon the discussion period is about to end. Please go over the response from the authors and initiate discussion.

regards

AC

---

### Meta-Review · Area_Chair_QczK · 2024-12-21

**Metareview:**

Dear authors,

This draft received overall positive ratings. Please take in consideration input from the reviewers to update the draft.
Please not that reviewers have given good rating with the condition of draft to be updated e.g. "I would emphasize that in case this paper is accepted by AC decision, please take into account the above explanations to make major improvements in the method section, either writing part or coming up with a better figure for the main method if possible."
Other reviewer has asked the code to be released.

I concur their requests. Draft should be updated and full implementation  (covering all aspects, experiment settings, configs, etc..) should be made publicly available before the camera-ready version is submitted.

A side note:
* Abstract should be revised to be improve readability and clarity.
* ”vanilla model” appears to be defined in the Caption of Fig.1 (Here, the term ”vanilla model” refers to a model that employs
* no specialized techniques to address imbalanced data.). Since the term is being used later in the paper too, such definitions should be in the main draft.
* Suggestion: Remove "bold" format from "Thus, it is crucial to explore ...."
* Suggestion: Dist Loss is quite generic term, changing to something more specific will help remove any confusion.




regards

AC

**Additional Comments On Reviewer Discussion:**

Reviewers have updated their score after author feedback. Reviewers requested release of code and inclusion of information presented during feedback into the draft.

---

### Decision · Program_Chairs · 2025-01-22

Accept (Poster)